# Homozygosity for a Rare *Plec* Variant Suggests a Contributory Role in Congenital Insensitivity to Pain

**DOI:** 10.3390/ijms25126358

**Published:** 2024-06-08

**Authors:** Piranit Kantaputra, Teerada Daroontum, Kantapong Kitiyamas, Panat Piyakhunakorn, Katsushige Kawasaki, Achara Sathienkijkanchai, Pornswan Wasant, Nithiwat Vatanavicharn, Thippawan Yasanga, Massupa Kaewgahya, Sissades Tongsima, Timothy C. Cox, Stefan T. Arold, Atsushi Ohazama, Chumpol Ngamphiw

**Affiliations:** 1Center of Excellence in Medical Genetics Research, Faculty of Dentistry, Chiang Mai University, Chiang Mai 50200, Thailand; shopkantapong@gmail.com (K.K.); kwgmsp@gmail.com (M.K.); 2Division of Pediatric Dentistry, Department of Orthodontics and Pediatric Dentistry, Faculty of Dentistry, Chiang Mai University, Chiang Mai 50200, Thailand; 3Department of Pathology, Faculty of Medicine, Chiang Mai University, Chiang Mai 50200, Thailand; mewteerada@gmail.com; 4Panare Hospital, Dental Public Health Division, Panare District, Surat Thani 94130, Thailand; send2gift@yahoo.com; 5Division of Oral Anatomy, Faculty of Dentistry & Graduate School of Medical and Dental Sciences, Niigata University, Niigata 950-2181, Japan; ka2shige@dent.niigata-u.ac.jp (K.K.); atsushiohazama@dent.niigata-u.ac.jp (A.O.); 6Division of Medical Genetics, Department of Pediatrics, Faculty of Medicine, Siriraj Hospital, Mahidol University, Bangkok 73170, Thailand; geneticsped@gmail.com (A.S.); pornswan.was@gmail.com (P.W.); nithiwat_v@hotmail.com (N.V.); 7Medical Science Research Equipment Center, Research Administration Section, Faculty of Medicine, Chiang Mai University, Chiang Mai 50200, Thailand; thippawan.y@cmu.ac.th; 8National Biobank of Thailand, National Center for Genetic Engineering and Biotechnology (BIOTEC), Pathum Thani 12120, Thailand; sissades.ton@biotec.or.th (S.T.); chumpol.nga@gmail.com (C.N.); 9Departments of Oral & Craniofacial Sciences, School of Dentistry, and Pediatrics, School of Medicine, University of Missouri-Kansas City, Kansas City, MO 64108, USA; coxtc@umkc.edu; 10Biological and Environmental Science and Engineering Division, Computational Bioscience Research Center, King Abdullah University of Science and Technology (KAUST), Thuwal 23955-6900, Saudi Arabia; stefan.arold@kaust.edu.sa

**Keywords:** absence of pain sensation, acro-osteolysis, lack of pain sensation, dental anomalies, tooth agenesis, corneal scar, *Plec* mutation

## Abstract

Congenital insensitivity to pain is a rare human condition in which affected individuals do not experience pain throughout their lives. This study aimed to identify the molecular etiology of congenital insensitivity to pain in two Thai patients. Clinical, radiographic, histopathologic, immunohistochemical, and molecular studies were performed. Patients were found to have congenital insensitivity to pain, self-mutilation, acro-osteolysis, cornea scars, reduced temperature sensation, tooth agenesis, root maldevelopment, and underdeveloped maxilla and mandible. The skin biopsies revealed fewer axons, decreased vimentin expression, and absent neurofilament expression, indicating lack of dermal nerves. Whole exome and Sanger sequencing identified a rare homozygous variant c.4039C>T; p.Arg1347Cys in the plakin domain of *Plec*, a cytolinker protein. This p.Arg1347Cys variant is in the spectrin repeat 9 region of the plakin domain, a region not previously found to harbor pathogenic missense variants in other plectinopathies. The substitution with a cysteine is expected to decrease the stability of the spectrin repeat 9 unit of the plakin domain. Whole mount in situ hybridization and an immunohistochemical study suggested that *Plec* is important for the development of maxilla and mandible, cornea, and distal phalanges. Additionally, the presence of dental anomalies in these patients further supports the potential involvement of *Plec* in tooth development. This is the first report showing the association between the *Plec* variant and congenital insensitivity to pain in humans.

## 1. Introduction

Pain is a defense mechanism mediated by nociceptors that warn living organisms of noxious or harmful stimuli [1]. It is widely recognized as playing a crucial role in safeguarding human health [2]. Nociceptor neurons express receptors to detect harmful or noxious stimuli, including cold, heat, reactive chemicals, and physical traumas [1,3]. Genetic disorders that may result in the absence of pain sensation include congenital insensitivity to pain, hereditary sensory neuropathy, and hereditary sensory and autonomic neuropathy (when the autonomic nervous system is involved) [4].

Congenital insensitivity to pain, also known as congenital painlessness, is a rare human phenotype in which affected individuals do not experience pain throughout their lives [5]. This condition is typically present from birth but may not be recognized by parents until later in life. However, it is important to note that individuals with hereditary sensory and autonomic neuropathy may experience a gradual loss of pain sensitivity over time [6]. Individuals with congenital insensitivity to pain typically display a number of distinctive features, such as the lack of pain perception, decreased sensitivity to temperature, increased vulnerability to *Staphylococcus aureus* infection, the loss of the sense of smell, urinary incontinence, and delayed pupillary light reflexes. Furthermore, they may exhibit corneal scars, acro-osteolysis, and Charcot joints [5]. The absence of pain sensation can have serious consequences, as pain plays a vital role in alerting us to potential harm and teaching us how to use our bodies safely and effectively [5]. Congenital insensitivity to pain may result in repeated self-injury and fractures, which can lead to mutilation, amputation, and impaired healing [6].

Congenital insensitivity to pain may be an isolated symptom or manifest as a prominent feature among other hereditary sensory and autonomic neuropathies. Several genes have been reported to be associated with congenital insensitivity to pain or hereditary sensory and autonomic neuropathies. Of all those genes, *NTRK1* and *SCN9A* are the most commonly involved [6,7]. Genetic alterations in these genes have been shown to result in developmental disorders of pain-sensing neurons, the altered electrical activity of nociceptors, or the neurodegeneration of peripheral nerves [6].

*Plectin* is a 4684 residue multidomain and multifunctional cytolinker protein and signaling scaffold belonging to the plakin protein family. *Plectin* orchestrates the structural and functional organization of filamentous cytoskeletal networks [8], functioning as a key cytoskeleton molecule facilitating crosstalk between intermediate filaments, microtubules, and actinomyosin. It is expressed abundantly in both the central and peripheral nervous systems, playing a critical role in both systems through its interactions with the various cytoskeletons, anchoring proteins, plasma membrane receptors, extracellular matrix, and enzymes involved in intracellular signaling [9], particularly affecting processes in the brain and spinal cord [10].

Genetic variants in human *Plec* have been reported to be associated with a number of *Plectin*-associated diseases (plectinopathies), including autosomal recessive generalized intermediate epidermolysis bullosa simplex 5D (EBS5D; MIM 601487), autosomal dominant epidermolysis bullosa simplex 5A, Ogna type (EBS5A; MIM 131950), autosomal recessive epidermolysis bullosa simplex 5B with muscular dystrophy (EBS5B; MIM 226670), autosomal recessive epidermolysis bullosa simplex with pyrolic atresia (EBSPA; MIM 612138), and autosomal recessive limb-girdle muscular dystrophy-17 (LGMDR17; MIM 613723) [9,10,11]. In the majority of cases, the underlying genetic changes in these conditions are truncating or indel variants in one or both alleles. These presumed loss of function (LOF) variants have been reported throughout the gene. A small number of missense variants have been reported either on one allele with an LOF variant on the other allele, or more rarely as compound heterozygous missense or homozygous missense variants, with these located, with few exceptions, in the C-terminal rod domain.

Here, we report two consanguineous Thai siblings affected with congenital insensitivity to pain, self-mutilation, acro-osteolysis, cornea scars, reduced temperature sensation, tooth agenesis, root maldevelopment, and underdeveloped maxilla and mandible. A rare homozygous missense variant in the *Plec* gene was found to be associated with the phenotypes. The variant changes a conserved residue at the end of the N-terminal plakin domain of the protein, a domain in which few missense variants have been reported in patients [11]. This appears to be the first report of congenital insensitivity to pain caused by a unique genetic variant in the *Plec* gene. Furthermore, our study highlights an association between a *Plec* variant and dental anomalies, which has not been previously reported.

## 2. Results

### 2.1. Patient Reports

#### 2.1.1. Patient 1

Patient 1, a 28-year-old male, has been unable to feel pain since birth. However, his parents only became aware of this when he was 1.5 years old. From the age of 6, he has experienced chronic inflammation and infection with pustules at the tips of his fingers and toes. This presentation suggested multiple episodes of *Staphylococcus aureus* infections of the skin, which resulted in auto-amputation (Figure 1A,B and Figure 2A,B). At the age of 9, a radiographic examination of the hands revealed delayed bone age and acro-osteolysis of the fingers, while the thumbs showed no signs of acro-osteolysis. Furthermore, a radiographic examination of the feet showed severe acro-osteolysis with the loss of the distal and middle phalanges of the toes, but the great toes were not affected (Figure 2C,D).

The patient complained that his teeth were loose upon eruption and were very annoying. Some teeth fell out by themselves, but some were pulled by the patient. At the age of 9, an oral examination revealed the presence of three permanent teeth with no primary teeth present. The patient exhibited self-mutilation activities, specifically lip and tongue biting. Additionally, oral constriction resulting from scar contraction was observed (Figure 1A,B). Radiographs of the lateral skull and postero-anterior skull taken at 9 years old revealed the severe underdevelopment of the maxilla and mandible (Figure 1C,D). The panoramic radiograph showed that the maxilla and mandible were underdeveloped, and that three permanent teeth had erupted. Furthermore, six developing permanent teeth were observed, which suggests the agenesis of multiple permanent teeth (Figure 1E).

At the age of 28, the patient presented with multiple facial scars, a bulbous nose, and oral constriction (Figure 3A). The patient’s sweating and lacrimation were normal, but thermal sensation was reduced. Additionally, the patient reported a history of self-injury while consuming hot food. The patient did not display any additional autonomic dysfunctions, such as anhidrosis, frequent fever, blood pressure fluctuation, or gastrointestinal disturbances. Furthermore, the patient’s intelligence and touch sensation were within normal ranges (i.e., the patient could feel the bite of a mosquito). Upon examination, it was discovered that there was a loss of distal tips of fingers and toes, with multiple scars present from self-mutilation. Interestingly, the thumbs showed no signs of a loss of the tips (Figure 3C,E). Radiographic examination revealed acro-osteolysis of fingers and toes, with the loss of the middle and distal phalanges (Figure 3E,F). The patient experienced tooth loss as a result of autoextraction and dental caries. A panoramic radiograph taken at the age of 28 showed underdeveloped maxilla and mandible as well as the absence of multiple permanent teeth (Figure 3G).

The patient’s vision was described as blurry, such that he required assistance for many tasks, such as eating and walking outside the house. Corneal opacification was first observed at the age of 6 months, and the patient had no memory of ever having eyesight. An ophthalmological examination at the age of 28 confirmed blindness in both eyes. Both corneas were severely clouded. The evaluation of the optic nerves was hindered by the presence of large corneal scars (Figure 3A,B). At the age of 9, the electroretinogram (ERG) revealed generalized abnormal retinal function; however, the visual evoked potential (VEP) showed normal optic nerve and macular function. The fundus images of the right and left eyes showed unremarkable optic nerves and maculae.

#### 2.1.2. Patient 2

Patient 2, the younger brother of patient 1, was 19 years old at the time of his most recent examination. Congenital insensitivity to pain was recognized by the parents at the age of 6 months (Figure 4A,B). He was allergic to the antibiotic Cloxacillin. Sweating and lacrimation were unremarkable. The patient presented with reduced thermal sensation, while intelligence and touch sensation were normal (Figure 5A). Additionally, the patient had a history of chronic inflammation and infection with pustules at the tips of his fingers and toes, suggestive of chronic *Staphylococcus aureus* infections. This was followed by auto-amputation (Figure 5C,D). A radiographic examination of the hands showed acro-osteolysis at the tips of the second distal phalanges. The radiograph of both feet indicates significant acro-osteolysis of the middle and distal phalanges of the toes (Figure 5E,F).

At 9 months of age, the patient displayed self-mutilation activities, such as lip and tongue biting (Figure 4A,B). By the age of 19, nine permanent teeth had erupted. The patient reported that some teeth were loose upon eruption, which caused discomfort. The patient removed some of his own teeth, while others fell out on their own. More recently, the mandibular right second premolar was extracted by a dentist using local anesthesia. Scar contracture around the mouth resulting from lip biting was observed (Figure 5A). At the age of 19, a panoramic radiograph revealed severely underdeveloped (narrow) maxilla and mandible, with only nine permanent teeth present, indicating the agenesis of multiple teeth (Figure 5G). It was also noted that the maxillary right permanent first molar was single-rooted, and the maxillary second permanent molars were taurodontic.

At the age of 1, he had herpes keratitis in both eyes, which finally resulted in a corneal ulcer or opacity in his right eye by the age of 5. An ophthalmological examination at the age of 19 revealed that he had limited visual acuity (Figure 5B).

### 2.2. Whole Exome Sequencing and Bioinformatic Analysis

Whole exome and Sanger sequencing identified a homozygous *Plec* variant (NM_000445.3: c.4039C>T; NP_000436.2: p.Arg1347Cys; chr8:145001038G>A based on human reference sequence build GRCh37 (hg19); rs372256096) in patients 1 and 2. The individuals I-1 and I-3 carried the wild-type alleles. The other unaffected family members were heterozygous for the variant (Figure 1 and Figure 8A). Of note, the patients’ family and the individual I-4 were unrelated but have lived in the same village for generations. Therefore, a founder effect is suspected. The c.4039C>T: p.Arg1347Cys variant is predicted to be deleterious (0.07), probably damaging (1.0), disease probability (0.832), and deleterious (-7.029) by SIFT (https://sift.bii.a-star.edu.sg/: accessed on 8 January 2024), PolyPhen-2 (http://genetics.bwh.harvard.edu/pph2/: accessed on 8 January 2024), SNPs&GO (http://snps.biofold.org/snps-and-go//snps-and-go.html: accessed on 8 January 2024), and PROVEAN (http://provean.jcvi.org/index.php: accessed on 8 January 2024), respectively. It is predicted as most likely to interfere with protein function (C65) and altered transmembrane protein (0.12) by Align-GVGD (http://agvgd.hci.utah.edu/: accessed on 8 January 2024) and MutPred (http://mutpred.mutdb.org/index.html: accessed on 8 January 2024), respectively. Additionally, the CADD (https://cadd.gs.washington.edu/: accessed on 8 January 2024) score of this variant is 24.8. The amino acid is conserved across species (Figure 6B). The bioinformation analysis revealed the absence of the variants in the known congenital insensitivity to pain genes, including *ATL1*, *ATL3*, *GLA*, *KIF1A*, *NGF*, *DST*, *ELP1*, *PRDM12*, *RAB7A*, *RETREG1*, *ZFHX2*, *SPTLC2*, *TTR*, *WNK1*, *SCN11A*, *SPTLC1*, *MPV17*, *NAGLU*, *CLTCL1*, *FAAHP1*, *FLVCR1*, *SCN9A*, and *NTRK1* [6,7]. We have searched for rare heterozygous and homozygous variants in these genes and considered all possible modes of inheritance but have not found any other possible candidate variants for the congenital insensitivity to pain phenotype.

### 2.3. Histopathological Study of Skin Biopsy

Histopathological studies of the skin biopsies of patients 1 and 2 were similar. The epidermis appeared to be of normal thickness. Papillary and reticular dermis were unremarkable. Sebaceous glands, eccrine sweat glands, and superficial vascular plexus appeared unremarkable. No nerve bundles were observed in comparison to normal skin (Figure 7A–C).

### 2.4. Immunohistochemistry of the Skin

The immunohistochemical localization of vimentin and neurofilament in representative samples of normal skin (control) as well as in patient 1 and patient 2 samples was performed. Vimentin is expressed in the cytoplasm of mesenchymal cells, including melanocytes, endothelial cells, fibroblasts, and vascular smooth muscles cells. The normal skin control exhibited stronger vimentin staining than those of patients 1 and 2 (Figure 8A,C,E). The immunohistochemical localization of neurofilament was detected in the dermal nerves of the normal (control) skin but was not detected in the skin biopsies of both patients (Figure 8B,D,F).

### 2.5. Transmission Electron Microscopic Study of Skin

The transmission electron microscopic study of the representative sample of the skin of patient 1 showed a fewer number of axons than normal (Figure 9A,B).

### 2.6. Plec Expression during Early Development

#### 2.6.1. *Plectin* Expression in the Spinal Dorsal Horn

To investigate the potential involvement of *Plectin* in the congenital insensitivity to pain observed in our patients, an analysis of *Plectin* expression in the spinal dorsal horn of mouse embryos at E18.5 was conducted. Double staining with a neuronal cell marker, microtubule-associated protein 2 (MAP2), revealed that *Plectin* expression overlapped with MAP2 expression in the dorsal horn, indicating that *Plectin* is expressed in spinal dorsal horn neurons (Figure 10A–H).

#### 2.6.2. Underdeveloped Jaws and Corneal Defects

Both patients with the homozygous *Plec* variant presented with underdeveloped maxillae and mandibles as well as corneal ulcerations and scars, suggesting an important role for *Plec* in the development of these structures. In support of this, the whole mount in situ hybridization of *Plectin* at embryonic day (E) 10.5 and an immunohistochemical analysis of the developing cornea of mouse embryos were performed. *Plectin* was found to be strongly expressed in developing maxillary and mandibular processes and in the developing cornea (Figure 11A,B).

#### 2.6.3. *Plec* Expression in the Developing Distal Phalanges

Acro-osteolysis and chronic infections at the tips of the fingers are the major findings found in our patients. In order to better understand the development of distal digits, we conducted an immunohistochemical study of mouse embryo digits. From E13.5, the anlagen of the distal phalanges in mice were observed as condensed mesenchyme [12,13]. Furthermore, we were able to detect *Plectin* expression in the condensed mesenchyme at E14.5 (Figure 12A,B). At E16.5, it was observed that cartilage was present in the distal phalange primordia, while condensed mesenchyme surrounded it. It is worth noting that *Plectin* expression was not detected in the cartilage of the distal phalange primordia, but it was found in the surrounding condensed mesenchyme (Figure 12C,D). Ossification is initiated at the distal tip of the distal phalange around E18.5. *Plectin* was expressed in the distal tip of the distal phalange at postnatal (P) day 0 (Figure 12E,F). The restricted expression of *Plectin* was also observed in the distal tip of the connective tissue (Figure 12E,F).

### 2.7. Mutant Protein Model

This p.Arg1347Cys variant is located in the SR9 region of the plakin domain (Figure 13A). The SR7-SR9 region has been shown to be a rigid unit resistant to mechanical deformation, and this function seems to be important for the normal function of *Plectin*/plakins [14]. The amino acid residue Arg1347 is located on the three helical bundle structures of SR9 (Figure 13B), where it is likely stabilizing the structure through polar contacts with nearby residues Y1343, E1293, and Q1296 (Figure 13C). The replacement of Arg1347 with a cysteine results in the loss of these stabilizing bonds and reducing the stability of the SR9 unit. Furthermore, the positively charged Arg1347 is part of an extended basic surface patch, which would be diminished by a cysteine in this position (Figure 13C,D). Although no specific protein–protein interactions have been identified within this region, they cannot be completely ruled out [14]. Interestingly, AlphaFold strongly suggests that the fragment SR7-9 dimerizes (ipTM = 0.76), implicating Arg1347 in dimer-stabilizing ionic and hydrogen bonds. These interactions would be lost in the cysteine variant. In support, the asymmetric unit of the SR7-9 crystal structure (PDB id 5J1I) contains the dimeric arrangement predicted by AlphaFold [14]. SR9 is positioned just before the rod domain, which has also been shown to homodimerize [15]. Hence, the p.Arg1347Cys variant would affect intra- and intermolecular interactions.

## 3. Materials and Methods

This study was conducted in accordance with the Declaration of Helsinki and national guidelines. Informed consent was obtained from the parents in accordance with the regulations of the Human Experimentation Committee of the Faculty of Dentistry, Chiang Mai University (certificate of approval number 25/2020). Clinical and radiographic examinations were performed on two sons (patients 1 and 2) of consanguineous Thai parents (Figure 14). Skin biopsies were taken for histological studies, transmission electron microscopy, and immunohistochemical investigations. Genetic variations were identified using Whole exome and Sanger sequencing.

### 3.1. Whole Exome Sequencing and Bioinformatic Analysis

The genomic DNA of the proband (patient 1; III-2), their parents (II-1 and II-2), and an unaffected male sibling (III-3) was sequenced using exome sequencing services from Macrogen in Seoul, Republic of Korea. This approach allowed for the investigation of all exonic regions, capturing the full spectrum of coding variation within the family. The capture library kit, SureSelect V6 UTR (Agilent Technologies, Santa Clara, CA, USA), was used to target all exonic regions as well as their corresponding untranslated regions. The sequencing data output in FASTQ format was processed using GATK4 best practices [16] to identify germline short variants based on the human reference sequence build GRCh38. The variants called for all family members (originally containing 673,593 variants) were combined to identify genotypes, resulting in a comprehensive genetic profile of the family.

Subsequently, the GENMOD tool [17] was used to generate possible modes of inheritance. In order to test the hypothesis of the autosomal recessive model, 6421 variants were extracted from the 177,299 variants that fitted the model. It should be noted that these variants were filtered based on strict criteria, including a genotype quality (GQ) greater than 20 and a read depth (DP) greater than 10.

The variants that met the filtering criteria were assigned deleterious impact scores using VEP build110 [18] with the plugin dbNSFP v4.4a [19]. To refine our list of candidate variants, we cross-referenced them with public variant databases, such as 1000G, gnomADe, and gnomADg. We only considered variants with allele frequencies below 0.0005 as potential candidates for the observed phenotype, resulting in eight variants. The study utilized a rigorous and systematic approach to filter out variants, specifically BayesDel_noAF, MetaLR, MutationTaster, and PROVEAN, which are the top performing functional prediction scores from dbNSFP v4.4a [19]. A summary of the bioinformatic pipeline is presented in Figure 15. Based on these scores, only one candidate variant, c.4039C>T; p.Arg1347Cys in the *Plec* gene, was identified as deleterious. The variant was confirmed by Sanger sequencing on two affected individuals (patients 1 and 2) and five unaffected family members. To verify the presence of the predicted variants, Sanger sequencing of exon 31 of the *Plec* gene was performed using the forward primer TGC ATT GCC ACA GGA CTA TGA and the reverse primer AGG TGG TGA GAT GGA ACC CT.

### 3.2. Histological Study

The patients’ skin was biopsied and then promptly fixed in a 10% neutral-buffered formalin solution. Following preservation, the tissues underwent dehydration, clearing, and infiltration with paraffin wax before being embedded in paraffin blocks. The resulting paraffin-embedded tissues were then sectioned at a thickness of 3 µm for histopathological analysis and stained with hematoxylin and eosin (H&E) using standard histological laboratory methods. The specimens were evaluated for histopathologic characteristics by a dermatopathologist with extensive experience.

### 3.3. Immunohistochemistry of the Skin

Sections of formalin-fixed, paraffin-embedded skin biopsy tissue were cut into 3 μm thick sections and placed on Superfrost Plus slides. The slides were placed in a dry oven set at 60 °C for one hour to soften the paraffin and to enhance tissue attachment. The immunohistochemistry procedure was performed on a Ventana BenchMark ULTRA autostainer per the manufacturer’s protocol. To summarize, the sections were deparaffinized, rehydrated, and antigen recovered using CC1 (prediluted, pH 8.0) antigen retrieval solution (Ventana, Ventana Medical Systems, Oro Valley, AZ, USA). Subsequently, the sections were incubated with the primary antibodies at a dilution suggested by the manufacturer. The following primary antibodies were used: vimentin (monoclonal V9; DAKO, dilution 1:1500) and neurofilament (monoclonal 2F11; DAKO, ready to use). The Ultraview universal DAB IHC detection kit was utilized for the visualization process. The slides were subsequently counterstained with hematoxylin and bluing solution. Following a gentle cleaning, the slides were dehydrated in graded ethanol and xylene before being mounted on a microscope slide using mounting media.

### 3.4. Transmission Electron Microscopy of Skin Biopsy

The selected skin biopsies were first fixed in 2.5% glutaraldehyde in 0.1 M PBS for a minimum of 24 h at 4 °C. Following this, the samples were post-fixed in 1% osmium tetroxide in 0.1 M PBS for at least 1 h. The samples were then dehydrated using a series of increasing concentrations of ethanol (50%, 70%, 85%, 95%, and 100%) before being infiltrated and embedded in epoxy resin (Embed-812, Electron Microscope Science, Hatfield, PA, USA). Ultrathin sections were obtained using an ultra-microtome (Leica Microsystems Inc., Deerfield, IL, USA) with a diamond knife. The sections were then stained with 1% uranyl acetate in 50% ethanol and lead citrate. Finally, they were examined and photographed using a JEOL JEM-2100 Plus transmission electron microscope (Tokyo, Japan).

### 3.5. Immunohistochemistry of Plec Expression in Mouse Embryo

CD1 mice were dissected to separate their heads, limbs, and trunks. The separated parts were fixed with 4% paraformaldehyde, wax embedded, and serially sectioned at 7 µm. Limbs and trunks were decalcified using 0.5 M EDTA (pH 7.6) after fixation. The sections were then incubated with the *Plectin* antibody (Santa Cruz Biotechnology, Dallas, TX, USA) and MAP2 antibody (Proteintech, Rosemont, IL, USA) overnight at 4 °C. The primary antibody against *Plectin* was detected using the Tyramide signal amplification system (Parkin Elmer Life Science, Shelton, CT, USA; fluorescein). Additionally, goat anti-rabbit Alexa Fluor 5948 (Abcam, Cambridge, UK) was used as a secondary antibody to detect MAP2.

### 3.6. Mutant Protein Modeling

The SR7-SR9 structure of *Plectin* was modelled based on the crystal structure of SR7-9 (PDB id 5jli), in which missing side chains were completed by SwissModel [20]. Results were consistent with a structural prediction of SR7-9 by AlphaFold [21]. The analysis was carried out as previously described [22]. SR7-9 homodimers were predicted using the ColabFold version of AlphaFold2 [23].

## 4. Discussion

### 4.1. Genetic Variant in Plec and Congenital Insensitivity to Pain

Genetic variants in human *Plec* have been reported to be associated with a number of multisystem conditions (called plectinopathies). These plectinopathies are typically caused by truncating or indel variants (i.e., LOF variants) in one or both alleles. Missense variants have rarely been reported but are typically in patients that present with characteristic clinical presentations for known plectinopathies. Further, the small number of missense variants are often on one allele with an LOF variant on the other allele or more rarely as compound heterozygous missense or homozygous missense variants. Notably, the missense variants reported in the classic plectinopathies are located, with few exceptions, in the C-terminal rod domain, suggesting the disrupted function of the C-terminal domain is key to these phenotypes [11].

Here, we report a rare homozygous variant c.4039C>T; p.Arg1347Cys (rs372256096) in the *Plec* gene in two Thai children of a consanguineous family with a clinical presentation distinct from other known plectinopathies. Both male patients exhibit congenital insensitivity to pain with acro-osteolysis, underdeveloped maxillae and mandibles, reduced thermal sensation, self-mutilation, lip and tongue biting, tooth pulling, and abnormal corneas. Five other unaffected family members were found to be heterozygous for the variant, indicating the full co-segregation of the variant with the phenotype. The c.4039C>T; p.Arg1347Cys variant is not found in the normal Thai population database of 2184 alleles (T-REx database: https://trex.nbt.or.th; accessed on 29 February 2024). According to the gnomAD v2.1.1, the variant c.4039C>T; p.Arg1347Cys has a global allele frequency of 0.0004320 in the general population with no reports of homozygotes (accessed from https://gnomad.broadinstitute.org on 12 May 2024). However, the gnomAD v4.1.0. reported its global allele frequency of 0.0001848 in the general population with a report of two homozygotes (accessed from https://gnomad.broadinstitute.org on 12 May 2024). It is important to note that the patients included in gnomAD v4.1.0 were not all “normal” (https://gnomad.broadinstitute.org/stats, accessed on 16 February 2024). In fact, ~470,000 exomes from the UK Biobank, of which ~10% of samples are considered “controls”, were included (accessed from https://www.ukbiobank.ac.uk on 14 May 2024). The further investigation of the two individuals that are also homozygous for this *Plec* variant is warranted to confirm the phenotypic association with pain insensitivity and dental anomalies. However, if there are truly healthy homozygotes of this variant, it would suggest that the presentation of the congenital insensitivity to pain phenotype may depend on other factors, including epigenetic factors, environmental factors, and the genetic background of the patients.

Importantly, the location of the amino acid substitution is in an N-terminal domain of *Plec* and thus also distinct from rare missense variants reported in the other plectinopathies. Further supporting the homozygosity of this variant as the cause of the presentation in both individuals is that mice lacking *Plectin* have also been reported to exhibit impaired pain sensitivity [24].

### 4.2. Structure, Complexity, and Expression of Plectin 

*Plec* is broadly expressed in numerous tissues, although it is particularly abundant in epithelial and muscle cells and in the central nervous system [9]. The *Plec* gene encodes a number of mRNA and protein isoforms due to the differential use of first exons in various cell types and tissues, which can result in different binding affinities to intermediate filaments.

The major *Plectin* isoform consists of a central 200 nm rod domain, encoded by exon 31, flanked by a large N-terminal globular domain and C-terminal domain, which are encoded by exons 1-30 and exon 32, respectively. The N-terminal globular domain has a differentially spliced exon 1 followed by an actin-binding domain (ABD) comprising two calponin homology domains and a plakin domain that consists of nine spectrin repeats and an SH3 [11,25]. The N-terminus serves as a binding site for intermediate filaments, such as vimentin, microfilaments such as actin, hemidesmosomal proteins such as integrin α6β4 and COL17A1, and the neuromuscular acetylcholine receptor clustering protein known as rapsyn. The C-terminal domain is composed of six *Plectin* repeat molecules with a linker region in between each segment. This region serves as binding sites for other intermediate filament proteins, such as desmin, vimentin, and cytokeratins; glial fibrillary protein; and integrin α6β4 [9,11,25]. The amino acid residue, Arg1347, is located in the SR9 region of the N-terminal plakin domain and is highly conserved across vertebrate species, suggesting its importance for *Plec* structure and function. Moreover, according to the protein structural analysis, the p.Arg1346Cys mutant destabilizes intra- and intermolecular interactions. The biological role of the plakin domain is proposed to involve buffering mechanical stress and mechanosensing [15]. Both of these functions involve unfolding and refolding in response to tension and require stability and dynamics to be finely tuned [26]. Although the replacement of Arg1347 by a cysteine may not be sufficient to abolish the monomeric structure in vitro, the loss of stabilizing intra- and intermolecular contacts is expected to perturb the interactions and dynamics of SR9 under physiological mechanical stress. Indeed, most missense mutations in the SR7–SR9 region of desmoplakin that are associated with disease are predicted to cause structural destabilization [15].

### 4.3. Functions of Plectin

*Plectin* is vital to the structural and functional integrity of cells and tissues exposed to mechanical stress, such as muscle, skin, vasculature, and intestine [10]. In the skin, *Plectin* is preferentially localized to hemidesmosomes. It mediates keratinocyte mechanical stability through its linkage of the intermediate filament network to hemidesmosomes with the concomitant suppression of counteracting contractile actomyosin forces [10]. However, *Plectin* also acts as a universal recruiter and anchoring platform of cytoplasmic intermediate filament networks [10] by interacting with other cytoskeletal elements, such as microtubules and actin, to form compact oligomeric structures via self-association. This ability of *Plectin* as a dynamic organizer of intermediate filament networks and to interact with a large number of different interacting partners is the basis of its role as a scaffolding platform for proteins involved in signaling [27,28]. According to *Plectin*’s numerous isoforms, universal expression, broad interaction potential, and extensive functional repertoire, the malfunction of *Plectin* results in a number of multisystem human malformations, as previously described [10,28].

### 4.4. Plectin in Central and Peripheral Nervous Systems

The discovery of a homozygous *Plec* variant in our patients who lacked pain sensation suggests the important role of *Plec* in pain perception. Pain sensation involves both the central and peripheral nervous systems, with sensory signals transmitted from the periphery to the dorsal horn of the spinal cord via primary afferent fibers. When an individual senses a damaging stimulus, nerve action potentials are transmitted to nociceptor cells in the dorsal root ganglia, which receive input from peripheral tissues, such as the skin. The transmission of painful stimuli is believed to occur from the periphery to the spinal medulla and brainstem through small myelinated and unmyelinated A and C fibers [2].

*Plectin* interacts with distinct types of cytoskeletons, anchoring proteins, plasma membrane receptors, extracellular matrix, and enzymes involved in intracellular signaling in the central nervous system [9]. According to Valencia et al. (2021) [24], *Plectin* isoform c was found to be the primary isoform expressed in both the peripheral and central nervous systems [24]. *Plectin* isoform c has an important role in axonal microtubule dynamics via an isoform-specific interaction with tubulin [24]. The absence of *Plectin* isoform c in neurons has been found to cause changes in microtubule dynamics and an excessive association with tau proteins. These changes have been observed to affect various aspects of neuronal function, including neuritogenesis, neurite branching, growth cone morphology, and translocation and the directionality of the movement of mitochondria and vesicles. Additionally, it has been noted that this absence leads to a reduction in nerve conduction velocity in motor neurons and a shift from large diameter to small diameter axons [24,29]. Our immunohistochemical confirmation that *Plectin* is expressed in spinal dorsal horn neurons is thus consistent with a causal role for the homozygous *Plec* variants in our patients. However, in contrast to *Plectin*-deficient mice, which have been shown to exhibit impaired long-term memory and a reduced fear-conditioned memory, and in addition to impaired pain sensitivity [24], our patient did not exhibit impaired long-term memory or reduced fear-conditioned memory.

The histologic examination and transmission electron microscopy showed the absence of nerve bundles and the reduction of axons in the dermis of our patients, respectively. Immunohistochemical studies showed decreased vimentin and a lack of neurofilament expression. The decreased expression of vimentin and the lack of neurofilament expression in the skin of our patients as a result of altered *Plectin* protein may contribute to these phenotypes, as the proper formation of the neurofilament axon network is important for the establishment and maintenance of axon caliber and nerve conduction velocity [30]. Vimentin plays a critical role in regeneration and in the growth and differentiation of neurons [31], and vimentin expression precedes and transiently coincides with the expression of neurofilament proteins in developing neurons, indicating that vimentin intermediate filaments play an important role as a scaffold to facilitate the formation of mature intermediate filament networks [32].

### 4.5. Genetic Variant in Plec and Corneal Ulceration and Scarring

The ocular manifestations in patients with congenital insensitivity to pain include superficial punctate keratitis, neurotrophic keratopathy, corneal ulcers, corneal opacities, and dry eye syndrome [33,34,35,36]. Corneal ulceration found in our patients have been reported in a number of patients with congenital insensitivity to pain. The cornea is a highly innervated and sensitive organ in the body. It is innervated by small nerve fibers, including those with minimal myelination (Aδ fibers) or unmyelinated (C fibers). These nerves derive from the ophthalmic branch of the trigeminal nerve [37]. It has been reported that patients with congenital insensitivity to pain may exhibit the absence of the corneal reflex or corneal sensitivity [37,38,39,40,41,42,43,44] and are mostly likely due to the lack of central corneal nerves [33,37].

Corneal scars in patients with congenital insensitivity to pain may result from a lack of corneal sensitivity, impaired tear production, and self-mutilation due to the absence of pain sensation [37]. Additionally, our immunohistochemical study of *Plectin* in wild-type mice at E14.5 showed its expression in the developing cornea, which suggests a role for *Plectin* in corneal development. Abnormal corneas in patients with congenital insensitivity to pain may be developmental consequences of genetic alterations. However, self-mutilation or uncontrolled eye rubbing due to the absence of pain sensation may worsen the severity of corneal ulceration. It is noteworthy that some patients with an absent corneal reflex did not have corneal scars, supporting the idea of the corneal abnormality as a neurodevelopmental defect.

### 4.6. Genetic Variant in Plec, Underdeveloped Jaw Bones, and Autoextraction of Teeth

The autoextraction of teeth or self-tooth extraction found in our patients has frequently been reported in other patients with congenital insensitivity to pain [40,45,46,47,48,49,50,51,52,53,54,55,56]. Self-tooth pulling in patients with congenital insensitivity to pain is a behavior that has yet to be fully explained. In this regard, it is noteworthy that the basal bones of maxillae and mandibles of our patients were underdeveloped, with minimal alveolar bone, in line with the high expression of *Plectin* at embryonic day 10 (E10) in the maxillary and mandibular processes. This limited alveolar bone may explain why the teeth were reported to be loose once they erupted into the oral cavities. These loose teeth may therefore have simply been bothersome to them, prompting the self-pulling.

### 4.7. Genetic Variant in Plec and Acro-Osteolysis (Amputation of Digits)

Acro-osteolysis leading to amputation of digits is common in patients with congenital insensitivity to pain [19,40,44,45,49,50,51,57,58,59,60,61,62,63,64,65,66,67,68,69,70,71,72,73,74,75,76,77,78,79,80,81,82,83,84,85,86,87,88,89,90,91]. Acro-osteolysis has been reported to be associated with a number of genetic syndromes, including *NOTCH2*-associated Hajdu-Cheney syndrome (MIM 600275), *CTSK*-associated Pycnodysostosis (MIM 601105), *CTSC*-associated Haim-Munk syndrome (MIM 245010), *IFIH1*-associated Singleton–Merten syndrome 1 (MIM 182250), *COL3A1*-associated Ehlers–Danlos syndrome, vascular type (MIM 130050), *PDGFRB*-associated premature aging syndrome, Penttinen type (MIM 601812), *DST*-associated hereditary sensory and autonomic neuropathy, type VI (HSAN6; MIM 614653) [92], and *WNK1*-associated hereditary sensory and autonomic neuropathy, type II (MIM 201300) (https://omim.org: 8 January 2024).

It is worth noting that *Plectin* has been reported to interact with WNK1 [93] and DST [94,95]. Additionally, patients with genetic variants in *WNK1* and *DST* have been found to exhibit congenital insensitivity to pain with acro-osteolysis [96,97] or distal contracture [98,99,100]. *Plectin*, WNK1, and DST are thought to be important in pain sensation and digit development. Genetic variants in these genes appear to share pathogenetic pathways that can cause congenital insensitivity to pain and acro-osteolysis or distal contracture.

Notably, our immunohistochemical examination of *Plec* expression during the early development of mouse digits showed a high expression of *Plec* in the condensed mesenchyme at E14.5 and E16.5 but no expression in the cartilage of distal phalange primordia. Furthermore, *Plectin* was found to be expressed in the distal tips of the distal phalanges at postnatal day 0, indicating that the aberrant formation of distal digits due to disrupted *Plec* function may contribute to acro-osteolysis or the amputation of digits in patients with congenital insensitivity to pain. That thumbs were not affected in our patients may be due to differences in their developmental processes [101,102]. In addition, the position of the thumbs may make them less prone to self-mutilation.

### 4.8. Genetic Variant in Plec, Staphylococcus aureus and Pastuerella multocida Infections, and Delayed Wound Healing

Both patients who lacked pain sensation experienced multiple episodes of chronic *Staphylococcus aureus* infections of the skin. It has been shown that the nervous system communicates with the immune system to maintain immune homeostasis. Impaired communication between these systems can contribute to disease development and progression [103]. Additionally, nociceptors, which share similarities with immune cells, are involved in host–pathogen defense. Nociceptors express receptors such as formyl peptide receptors and Toll-like receptors, which are capable of directly sensing microbes [1]. It has been observed that patients with impaired nerve growth factor (NGF)-tropomyosin receptor kinase A (TRKA) signaling are more susceptible to *Staphylococcus aureus* infection. This susceptibility is believed to be due to disruptive NGF-TRKA signaling through an evolutionarily conserved pathway in macrophages, where NGFB is released in response to *Staphylococcus aureus* [5,104]. The similarity in symptoms between patients with *TRKA* variants and our patients with a *Plec* variant suggests that *Staphylococcus aureus* infection in our patients also involves NGF-TRKA signaling. Further investigation could be pursued in this potential avenue.

At the age of 15, patient 1 was hospitalized for a few weeks due to a rapid cellulitis at the site of an injury caused by a bite from his domestic cat. It is worth noting that the infection was caused by *Pasteurella multocida*, an immobile, anaerobic, Gram-negative coccobacillus fermenting bacterium belonging to the *Pasteurellaceae* family [105]. It is therefore possible that the homozygous loss of *Plec* also increased the patient’s susceptibility to *Pasteurella multocida* infection. The delayed wound healing seen in both patients, which might be a consequence of the resultant decrease in vimentin expression [106], could also increase the chance of infection.

### 4.9. Plec Variant and Its Clinical Phenotypes

The clinical phenotypes found in our patients carrying a homozygous variant in the *Plec* gene include congenital insensitivity to pain, corneal scarring, self-mutilation, the autoextraction of teeth, acro-osteolysis, and underdeveloped maxilla and mandible. The heterozygous variant c.4039C>T; p.Arg1347Cys in the *Plec* gene is rare. As mentioned above, genetic variants in *Plec* have been reported to be involved in *Plectin*-associated diseases, including autosomal recessive generalized intermediate epidermolysis bullosa simplex 5D, autosomal dominant epidermolysis bullosa simplex 5A, Ogna type, autosomal recessive epidermolysis bullosa simplex 5B with muscular dystrophy, autosomal recessive epidermolysis bullosa simplex with pyrolic atresia, and autosomal recessive limb-girdle muscular dystrophy-17 [9,10,11]. The location of the variant, epigenetic factors, modifying genes, and genetic background of the patients may account for the differences in phenotypes.

## 5. Conclusions

We report two Thai patients with congenital insensitivity to pain. The clinical and radiographic features of the patients included the congenital absence of pain sensation, self-mutilation, acro-osteolysis, corneal scars, decreased temperature sensation, tooth agenesis, root maldevelopment, and underdeveloped maxilla and mandible. Skin biopsies showed fewer axons, decreased vimentin expression, and an absence of dermal nerves. Whole exome and Sanger sequencing identified a rare homozygous variant, c.4039C>T; p.Arg1347Cys, which is located in the SR 9 region of the plakin domain, a region not previously found to harbor pathogenic missense variants in other Plectinopathies. [11]. Whole mount in situ hybridization and immunohistochemistry suggested that acro-osteolysis, corneal scars, dental anomalies, and underdeveloped maxilla and mandible in the patients are in part developmental defects. In addition, the presence of dental anomalies in these patients further supports the potential involvement of the *Plec* gene in tooth development. This is the first report showing a putative association between a *Plec* variant and congenital insensitivity to pain in humans. These findings suggest a genotype–phenotype correlation (domain-specific missense variants with distinct phenotypes), and thus, this may represent a unique Plectinopathy.

## Figures and Tables

**Figure 1 ijms-25-06358-f001:**
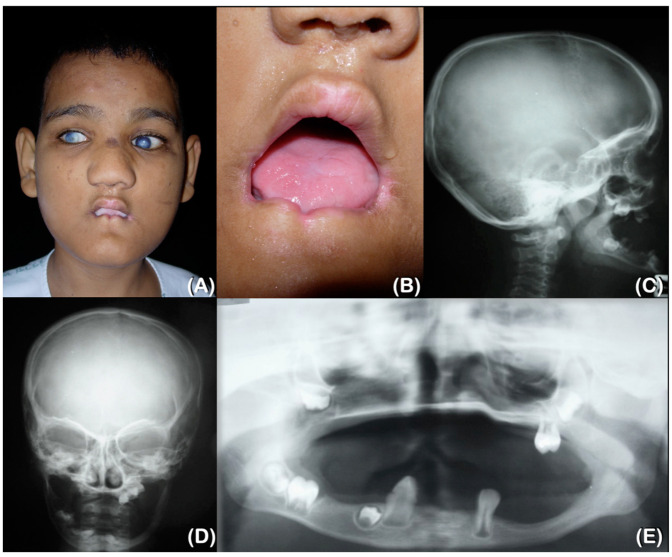
Patient 1 at 9 years of age. (**A**) Cloudy corneas. Bulbous nose. Microstomia. (**B**) Oral contracture and lip scars as a result of lip biting. (**C**,**D**) Lateral skull radiograph showing underdeveloped maxilla and mandible. (**E**) Three permanent teeth are erupted. Six developing permanent teeth are observed. The agenesis of multiple permanent teeth.

**Figure 2 ijms-25-06358-f002:**
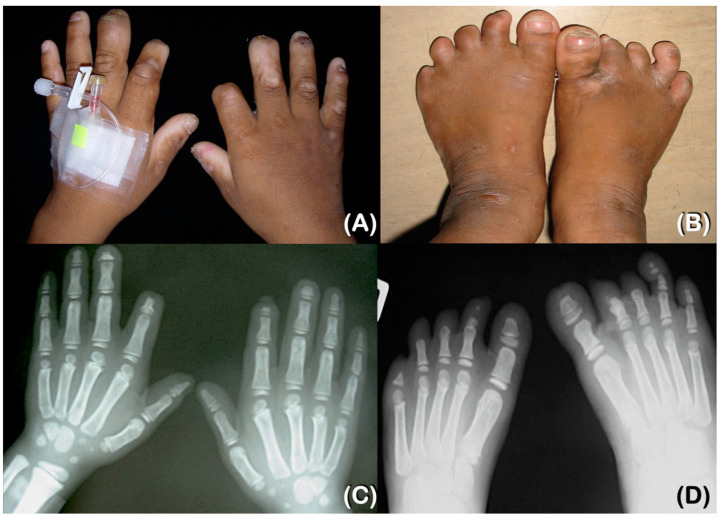
Patient 1 at 9 years of age. (**A**,**B**) Scars and the loss of the tips of fingers and toes. Chronic inflammation and infection with pustules at the tips of fingers and toes. (**C**) Hand radiograph showing delayed bone age and acro-osteolysis. Thumbs showing no signs of acro-osteolysis. (**D**) Foot radiograph showing severe acro-osteolysis with the loss of the distal and middle phalanges of the toes. Great toes are not affected with acro-osteolysis.

**Figure 3 ijms-25-06358-f003:**
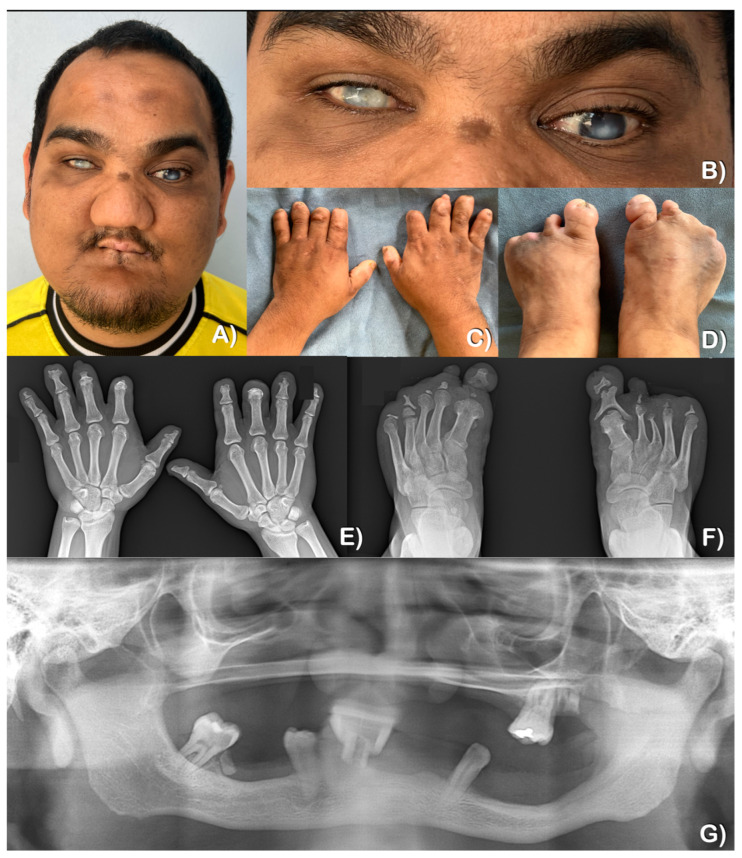
Patient 1 at 28 years of age. (**A**,**B**) Severe corneal scarring in both eyes. The right eye is more severely affected than the left one. Multiple facial scars, a bulbous nose, and an oral contracture with scars are noted. (**C**,**D**) Hands and feet. A loss of the distal tips of fingers and toes. Thumbs have scars from previous traumas. No signs of a loss of the tips. (**E**,F) Radiographs of hands and feet showing acro-osteolysis of fingers and toes. The loss of the middle and distal phalanges. (**G**) Panoramic radiograph at 28 years of age showing underdeveloped maxilla and mandible. The absence of multiple permanent teeth is observed.

**Figure 4 ijms-25-06358-f004:**
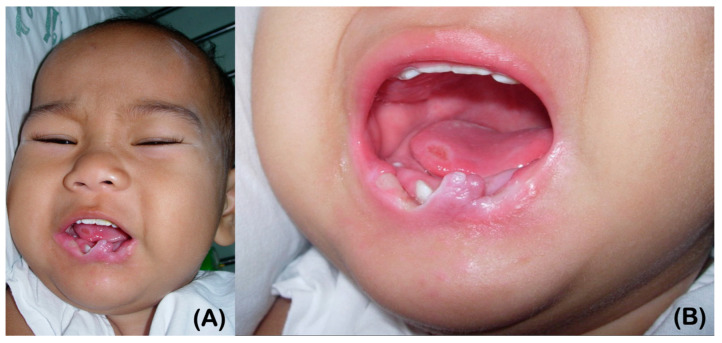
(**A**,**B**) Patient 2 at the age of 9 months. Ulcers as a result of lip and tongue biting are apparent.

**Figure 5 ijms-25-06358-f005:**
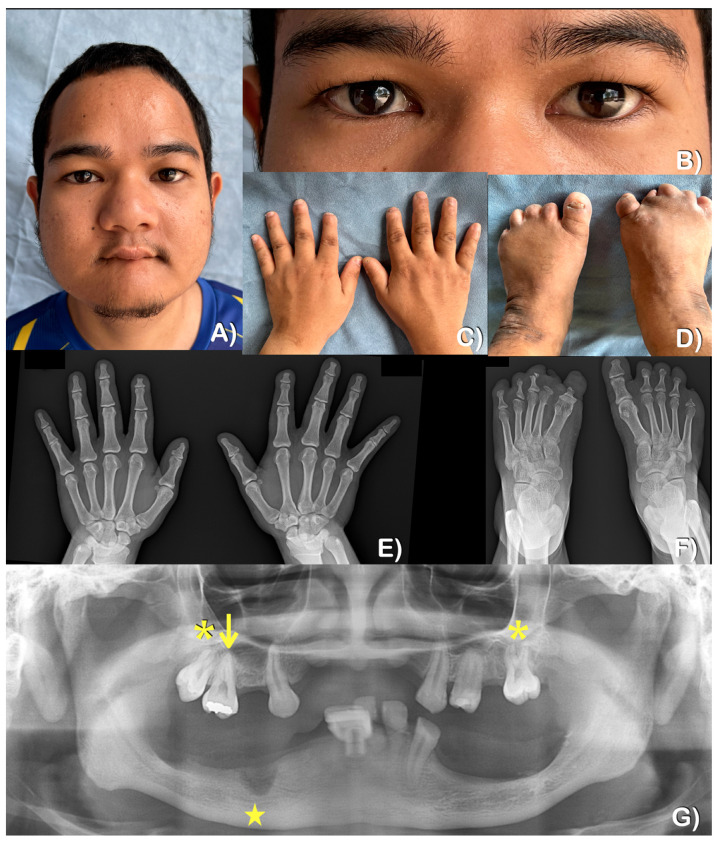
Patient 2 at the age of 19 years. (**A**) Corneal scar in the right eye. Oral constriction as a result of scar contracture. (**B**) Corneal scarring in the right eye. (**C**) The loss of the tips of the index fingers. Thumbs appear unremarkable. (**D**) The loss of the tips of all toes. Multiple scars are noted. (**E**) Hand radiograph showing acro-osteolysis of the tips of second distal phalanges. (**F**) Severe acro-osteolysis of the middle and distal phalanges of toes. (**G**) Panoramic radiograph showing underdeveloped (narrow) maxilla and mandible. The absence of most permanent teeth. Mild taurodontism at the left and right maxillary second permanent molars (asterisks). Unseparated roots of the maxillary right permanent first molar (arrow). Alveolar socket as a result of a recent tooth extraction (star).

**Figure 6 ijms-25-06358-f006:**
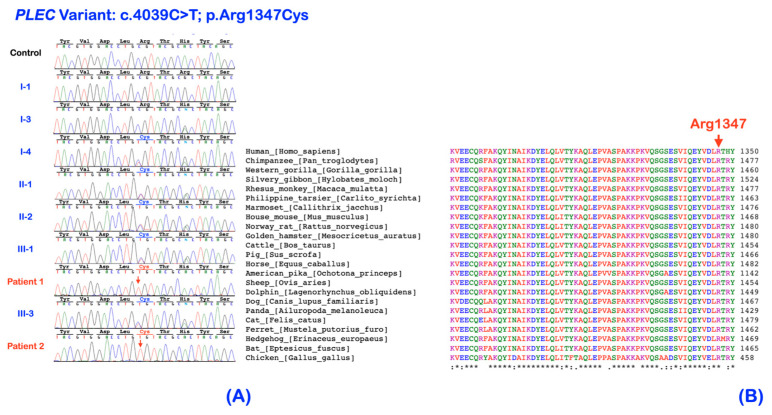
(**A**) Chromatograms of patients 1 and 2, their parents, and family members. Patients 1 and 2 are homozygous for the *Plec* variant (c.4039C>T; p.Arg1347Cys). The unaffected individual I-1 and I-3 have wild-type alleles. Other unaffected family members are heterozygous for the variant. (**B**) The conservation of amino acids in *Plectin* protein. The amino acid residue Arg1347 is conserved across species.

**Figure 7 ijms-25-06358-f007:**
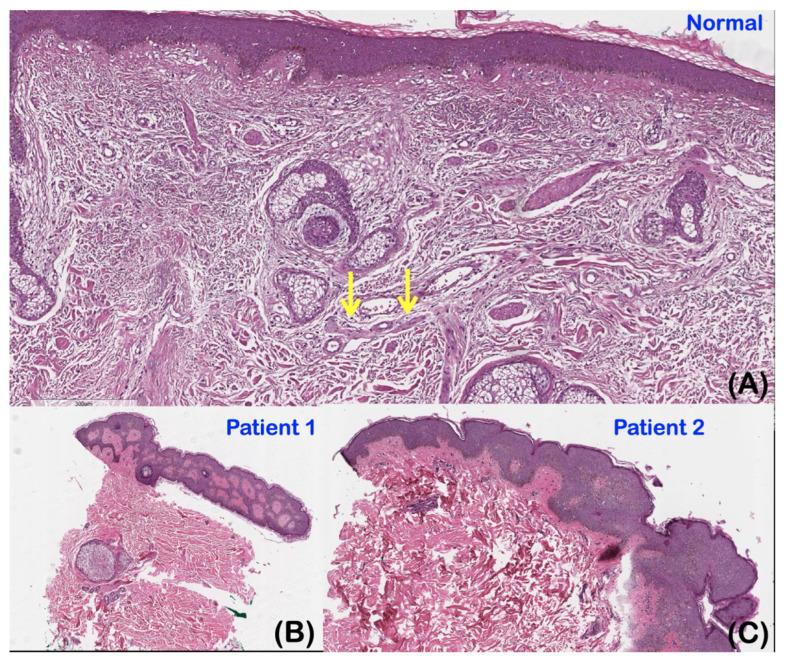
(**A**) Normal control. (**B**) Patient 1. (**C**) Patient 2. (**A**) The normal skin control showing a normal thickness of the epidermis. Its dermal skin appendages and dermal nerve (arrows) are unremarkable. (**B**,**C**) Patients’ epidermis showing normal thickness. Papillary and reticular dermis are unremarkable. Superficial vascular plexus is unremarkable. Nerve bundles are not seen. Sebaceous glands and eccrine sweat glands are unremarkable. (**C**) Mild superficial perivascular mononuclear cell infiltration.

**Figure 8 ijms-25-06358-f008:**
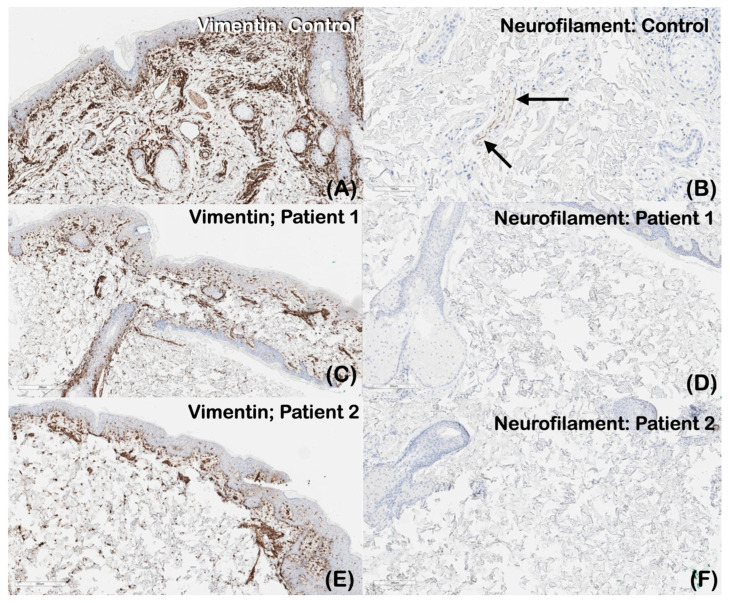
The immunohistochemical localization of vimentin in representative samples of (**A**) the normal skin control, (**C**) Patient 1, and (**E**) Patient 2. Vimentin is expressed in the cytoplasm of mesenchymal cells, including melanocytes, endothelial cells, fibroblasts, and vascular smooth muscles cells. The expression of vimentin in patients 1 and 2 is less than that of the normal skin control. The immunohistochemical localization of neurofilament in representative samples of (**B**) the normal skin control, (**D**) Patient 1, and (**F**) Patient 2. Neurofilament is expressed in the dermal nerves of the normal skin control (arrows) but is not found to be expressed in the skin biopsies of both patients.

**Figure 9 ijms-25-06358-f009:**
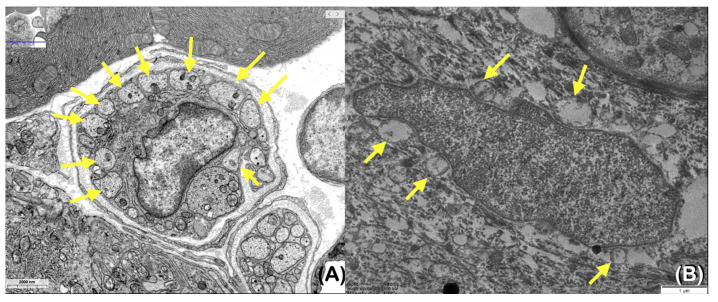
The transmission electron micrography of (**A**) normal skin. (**B**) The skin biopsy from patient 1 reveals a fewer number of axons (arrows) than those of (**A**) the normal skin (obtained from https://histologyguide.com/EM-view/RD-060-peripheral-nerve/06-photo-1.html: 8 January 2024.

**Figure 10 ijms-25-06358-f010:**
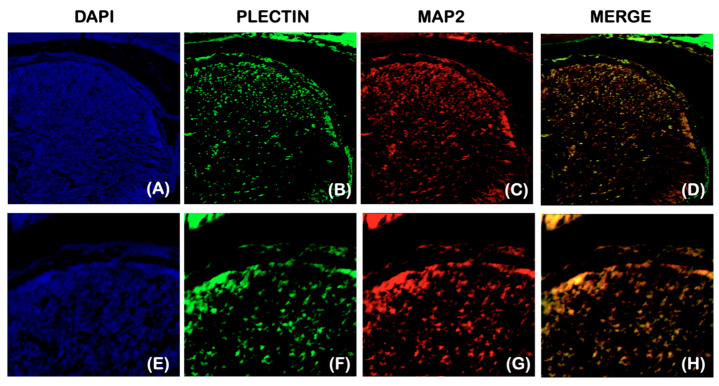
*Plectin* and MAP2 expression in the distal horn. Coronal sections showing the immunohisotochemistry of *Plectin*1 and MAP2 in wild-type mouse spinal dorsal horn. (**E**–**H**) are high magnifications of (**A**–**D**). *Plectin* expression is overlapped with MAP2 expression in the dorsal horn, suggesting that *Plectin* is expressed in spinal dorsal horn neurons.

**Figure 11 ijms-25-06358-f011:**
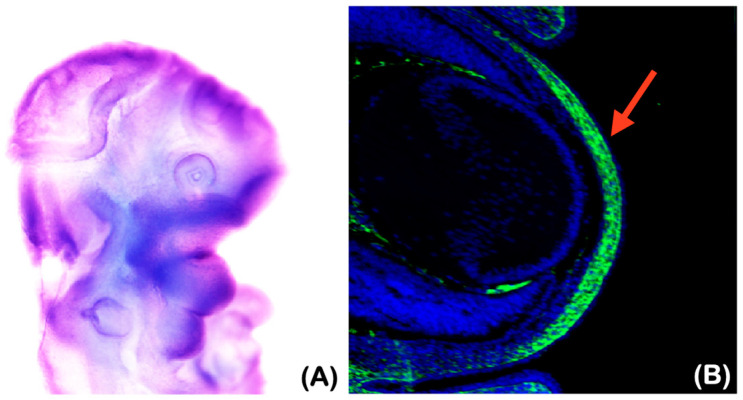
*Plectin* expression in the developing jaw and cornea. (**A**) A lateral view of the wild-type mouse head showing the whole mount in situ hybridization of *Plectin* at embryonic day (E) 10.5. Arrows indicating the expression of *Plectin* in the maxillary and mandibular processes. (**B**) Frontal section showing immunohisotochemistry of *Plectin* in wild-type mouse cornea at E14.5. Arrows indicating the expression of *Plectin* in developing cornea.

**Figure 12 ijms-25-06358-f012:**
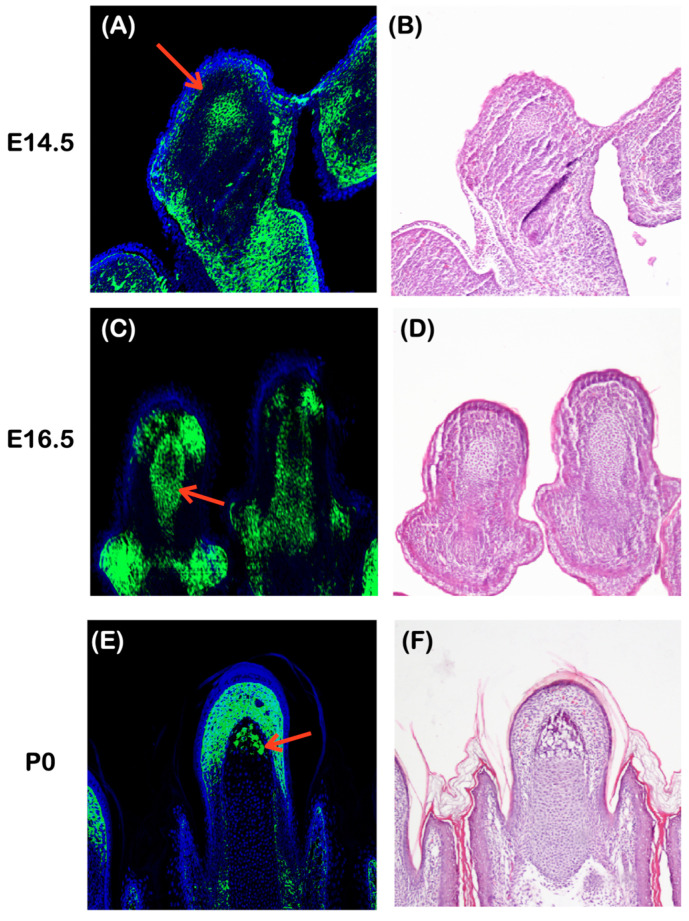
*Plectin* expression in digits. Coronal sections showing the immunohisotochemistry of *Plectin* (**A**,**C**,**E**) and morphology (**B**,**D**,**F**) in wild-type mouse digits at E14.5 (**A**,**B**), E16.5 (**C**,**D**), and postnatal (P) day 0 (**E**,**F**). (**B**,**D**,**F**) are adjacent sections of (**A**,**C**,**E**), respectively. Arrows indicating *Plectin* expression in condensed mesenchyme (**A**,**C**) and the area of ossification (**E**).

**Figure 13 ijms-25-06358-f013:**
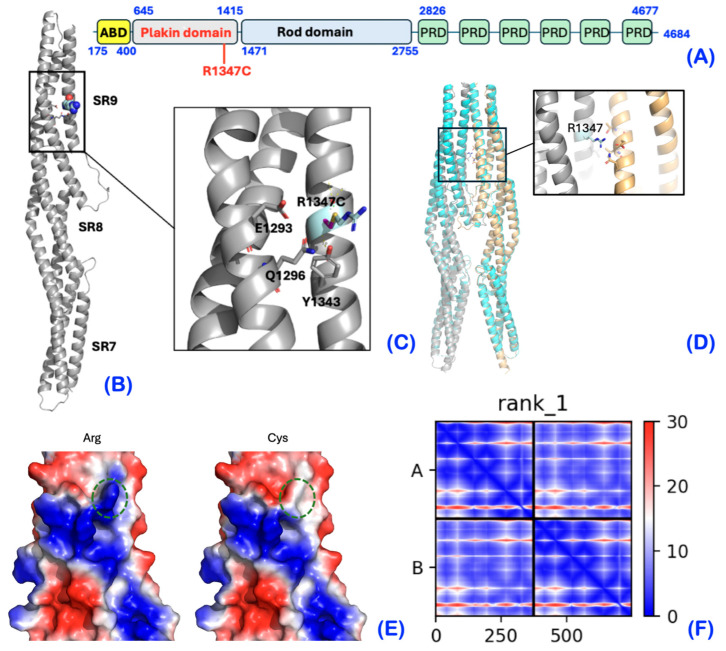
Mutant protein model. (**A**) Schematic drawing of human *Plectin* with domain boundary amino acids. SR: spectrin repeat; ABD: actin-binding domain, composed of two calponin homology domains; PRD: *Plectin* repeat domain. The location of the mutation Arg1347 in the ninth and last SR is indicated. The green square within the plakin domain indicates the SH3 domain. (**B**) The molecular model of the SR7-SR9 fragment based on PDB 5jli. Arg1347 is shown as a sphere model. (**C**) The zoom figure shows the molecular environment of Arg1347. The side chain of the cysteine is shown with carbons in magenta and the sulfur in yellow. The side chains of other key residues are highlighted. (**D**) AlphaFold predicts SR7-9 homodimers. The two protein chains of the AlphaFold model are colored in gray and pale yellow. The model is superimposed on the crystallographic dimer of SR7-9 (PDB id 5j1i, cyan). The boxed region containing Arg1347 is shown enlarged in the close-up figure. (**E**) The electrostatic surface of SR9, colored from blue (positively charged) over white (neutral) to red (negatively charged). Left and right views show the wild-type and mutant with position 1347 encircled (green). (**F**) Predicted aligned error (PAE) matrix for SR7-9 homodimerization.

**Figure 14 ijms-25-06358-f014:**
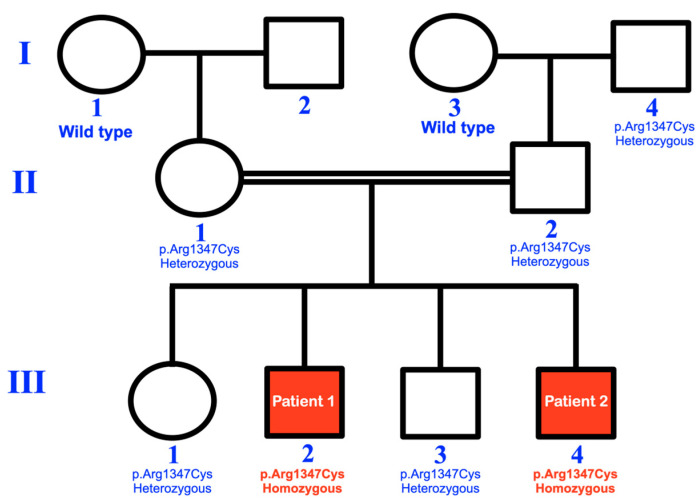
The pedigree of the consanguineous Thai family. The homozygous *Plec* variant c.4039C>T; p.Arg1347Cys variant is segregated with the congenital insensitivity to pain phenotype in two affected family members (III-2; patient 1 and III-4; patient 2). The individuals I-1 and I-3 have wild-type alleles. The other unaffected family members are heterozygous for the variant.

**Figure 15 ijms-25-06358-f015:**
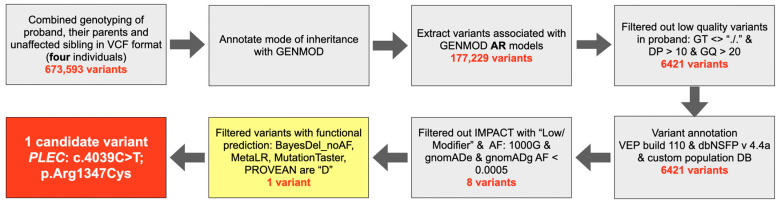
Whole exome sequencing and bioinformatics analysis to identify the candidate variant.

## Data Availability

The raw data supporting the conclusions of this article will be made available by the authors on request.

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
