# Peer review of "Homozygosity for a Rare Plec Variant Suggests a Contributory Role in Congenital Insensitivity to Pain"

_ijms, 2024, doi:10.3390/ijms25126358_

Round 1

Reviewer 1 Report

Comments and Suggestions for Authors

The manuscript ijms-3013504, "PLEC is a novel gene for congenital insensitivity to pain”, reports the identification of two Thai sibs presenting with congenital insensitivity to pain, self-mutilation, acro-osteolysis, corneal scars, reduced temperature sensation, tooth agenesis, root maldevelopment, and underdeveloped maxilla and mandible. The authors identified by whole exome sequencing a rare homozygous variant c.4039C>T; p.Arg1347Cys in the PLEC gene, which they associate with the clinical phenotype in the patients.

The results reported in the ms are indicative of an interesting finding and association between PLEC and congenital insensitivity to pain. However, with only two cases and very limited/weak data to support the pathogenic role of the identified missense variant, it's challenging to definitively establish that such a variant in PLEC is associated with congenital insensitivity to pain. Furthermore, mutations in the PLEC gene are associated with clinical phenotypes different from congenital insensitivity to pain.

This makes it difficult to establish causality.

The manuscript  has several flaws that need to be adjusted.

Here are some comments that can help the authors reconsider the findings.

Major comments:

With only two cases and a missense variant in the PLEC gene, implicated in other syndromes, it's challenging to definitively establish that such a variant is associated with congenital insensitivity to pain. Although there are supporting data, these are limited and weak, so it is difficult to draw broad conclusions about the gene's function and the mutation's impact. Therefore, the title "PLEC is a novel gene for congenital insensitivity to pain”, cannot be used.

The reported variant c.4039C>T; p.Arg1347Cys (rs372256096) for the selected transcript  NM_000445.3, is reported in Clinvar with conflicting classifications of pathogenicity, Uncertain significance(3); Benign(1); Likely benign(1), and in Uniprot as benign.

Although the findings from whole-mount in situ hybridization and immunohistochemistry support some of the clinical features associated with the phenotype of the two Thai patients, such as acro-osteolysis, corneal scars, dental anomalies, and underdeveloped maxilla and mandible, they are not enough to attest the potential involvement of PLECTIN in the congenital insensitivity to pain. The same is true for the mutant protein model. They are not strong supporting data.

In section 3.2. Whole exome sequencing and bioinformatic analysis, the authors stated that the variant rs37225609 is predicted to be disease-causing (0.999999827907939), deleterious (0.07), and probably damaging (1) by Mutation Tasting, SIFT, and PolyPhen-2. They should add references for these prediction tools. I could not find Mutation Tasting. Did the authors use Mutation Taster instead, as also stated in line 134 of the ms? In this last case, https://www.genecascade.org/MutationTaster2021/#transcript the variant c.4039C>T; p.Arg1347Cys (rs372256096) is predicted as benign not as disease-causing.

The reported variant is not located in a particular isoform of the PLEC gene or in an exon where mutations are not found in other associated phenotypes. The authors do not discuss enough the issue that this variant is associated with a clinical phenotype so different from the ones already described and do not give any possible explanation about it.

It seems the authors considered only the autosomal recessive model.

The authors stated that in order to test the hypothesis of the autosomal recessive model, 6,421 variants were extracted from the 177,299 variants that fitted the model and that the bioinformatic analysis revealed absence of the variants in the known congenital insensitivity to pain genes, including ATL1, ATL3, GLA, KIF1A, NGF, DST, ELP1, PRDM12, RAB7A, RETREG1, ZFHX2, SPTLC2, TTR, WNK1, SCN11A, SPTLC1, MPV17, NAGLU, CLTCL1, FAAHP1, FLVCR1, SCN9A, and NTRK1.

Considering the pathogenic role of the variant found in PLEC is not adequately support, and the clinical phenotypes associate with mutations in PLEC are different from the one reported in the two Thai patients, the authors, in case they have not already done, should check again their exome findings.

Are the known congenital insensitivity to pain genes adequately covered by probes in the SureSelect V6 UTR kit used?

Are there pathogenic/likely pathogenic variants in heterozygosity in these genes? Maybe the other allele is missing due to limitations of the technology, or maybe there is a different mechanism to be considered (i.e. see Complete paternal uniparental isodisomy for chromosome 1 revealed by mutation analyses of the TRKA (NTRK1) gene encoding a receptor tyrosine kinase for nerve growth factor in a patient with congenital insensitivity to pain with anhidrosis. Hum. Genet. 107: 205-209, 2000).

Did the authors look for a de novo event or a dominant model? I understand the two sibs come from a consanguineous marriage, but the autosomal recessive model is not the only one being considered.

They should have considered all these points before assuming the identified PLEC variant is associated with congenital insensitivity to pain.

Although not perfect, the Standards and Guidelines for the Interpretation of Sequence Variants (Richards et al., 2015), that I suggest using, do not indicate pathogenicity.

Minor comments

Figures 10, 12 and 16 should have a control to compare the findings.

In section 3.6.1. Plectin expression in the spinal dorsal horn, lines 343-345, “To investigate the potential involvement of PLECTIN in the congenital insensitivity to pain observed in our patients, an analysis of Plectin expression in the spinal dorsal horn of mouse embryos was conducted”. Please specify the day.

There are some inconsistencies for patient 2 in Fig. 6B and Fig. 8A, although the reported age is 19 for both figures, but the result is different, please adjust.

Figure 6. B) Patient 2. Corneal scar on the right eye. Left eye is unremarkable.

Figure 8. Patient 2 age 19 years. A) Corneal scars in both eyes

Furthermore, Fig. 6B can be moved to Fig. 8, so that there is one figure for each patient.

Author Response

Response to comments of Reviewer 1

            The manuscript ijms-3013504, "PLEC is a novel gene for congenital insensitivity to pain”reports the identification of two Thai sibs presenting with congenital insensitivity to pain, self-mutilation, acro-osteolysis, corneal scars, reduced temperature sensation, tooth agenesis, root maldevelopment, and underdeveloped maxilla and mandible. The authors identified by whole exome sequencing a rare homozygous variant c.4039C>T; p.Arg1347Cys in the PLEC gene, which they associate with the clinical phenotype in the patients.

  1. The results reported in the ms are indicative of an interesting finding and association between PLEC and congenital insensitivity to pain. However, with only two cases and very limited/weak data to support the pathogenic role of the identified missense variant, it's challenging to definitively establish that such a variant in PLEC is associated with congenital insensitivity to pain. Furthermore, mutations in the PLEC gene are associated with clinical phenotypes different from congenital insensitivity to pain. This makes it difficult to establish causality. The manuscript has several flaws that need to be adjusted. 

RESPONSE

Thank you for this comment. It is true that the previously reported mutations in PLEC are associated with clinical phenotypes different from congenital insensitivity to pain as we described in the introduction as follows.

Genetic variants in human PLEC have been reported to be associated with a number of Plectin-associated diseases (plectinopathies), including autosomal recessive generalized intermediate epidermolysis bullosa simplex 5D (EBS5D; MIM 601487), autosomal dominant epidermolysis bullosa simplex 5A, Ogna type (EBS5A; MIM 131950), autosomal recessive epidermolysis bullosa simplex 5B, with muscular dystrophy (EBS5B; MIM 226670), autosomal recessive epidermolysis bullosa simplex with pyrolic atresia (EBSPA; MIM 612138), and autosomal recessive limb-girdle muscular dystrophy-17 (LGMDR17; MIM 613723) [9-11]. In the majority of cases, the underlying genetic changes in these conditions are truncating or indel variants in one or both alleles. These presumed loss of function (LOF) variants have been reported throughout the gene. A small number of missense variants have been reported either on one allele with a LOF variant on the other allele, or more rarely as compound heterozygous missense or homozygous missense variants, with these located, with few exceptions, in the C-terminal rod domain.

PLECTIN is a multifunctional cytolinker protein and signaling scaffold belonging to the plakin protein family. Plectin orchestrates the structural and functional organization of filamentous cytoskeletal networks [Winter et al., 2016] and is a fundamental structure of all vital tissues of our body. To date, mutations in PLEC have been reported to cause clinical manifestations in muscle, skin, nails, and hair. However, it is noteworthy that mutations in PLEC can also cause malformations of the scarring anterior commissure of the larynx, urethral strictures of the bladder in patients with autosomal recessive generalized intermediate epidermolysis bullosa simplex 5D (https://omim.org/clinicalSynopsis/226670) or pyloric atresia and joint contractures in epidermolysis bullosa simplex 5C with pyloric atresia (MIM 612138; https://omim.org/entry/612138). Maldevelopment of the central nervous system has been reported in patients with PLEC-associated autosomal recessive limb-girdle muscular dystrophy-17 (LGMDR17; MIM 613723) as delayed motor development (https://omim.org/clinicalSynopsis/613723. Therefore, abnormal pain sensation or congenital insensitivity to pain is a possible symptom caused by a genetic variant in the PLEC gene.

It is important to note that the majority of reported mutations are LOF variants, with the few missense variants mostly in the C-terminal  rod domain - with patients identified because they share a phenotype with those having LOF variants. This new variant is a rare missense variant, not seen in the homozygous state in the normal population, that resides in the N-terminal plakin domain (spectrin 9 repeat) where no other missense variants have been described. We therefore propose that the specific disruption of the plakin domain is causing this distinct clinical presentation, whereas disruptions to the rod domain cause many of the other phenotypes). We apologize for not mentioning this in the first submission. This is also strongly supported by the study of Plectin-deficient mice, which have been shown to have impaired pain sensitivity (Valencia et al., 2021

Valencia, R.G.; Mihailovska, E.; Winter, L.; Bauer, K.; Fischer, I.; Walko, G.; Jorgacevski, J.; Potokar, M.; Zorec, R.; Wiche, G. Plectin dysfunction in neurons leads to tau accumulation on microtubules affecting neuritogenesis, organelle trafficking, pain sensitivity and memory. Neuropathol Appl Neurobiol 2021, 47, 73-95, doi:10.1111/nan.12635.

Here are some comments that can help the authors reconsider the findings.

Major comments:

  1. With only two cases and a missense variant in the PLEC gene, implicated in other syndromes, it's challenging to definitively establish that such a variant is associated with congenital insensitivity to pain. Although there are supporting data, these are limited and weak, so it is difficult to draw broad conclusions about the gene's function and the mutation's impact. Therefore, the title "PLEC is a novel gene for congenital insensitivity to pain”, cannot be used.

The reported variant c.4039C>T; p.Arg1347Cys (rs372256096) for the selected transcript  NM_000445.3, is reported in Clinvar with conflicting classifications of pathogenicity, Uncertain significance(3); Benign(1); Likely benign(1), and in Uniprot as benign. Although the findings from whole mount in situ hybridization and immunohistochemistry support some of the clinical features associated with the phenotype of the two Thai patients, such as acro-osteolysis, corneal scars, dental anomalies, and underdeveloped maxilla and mandible, they are not enough to attest the potential involvement of PLECTIN in the congenital insensitivity to pain. The same is true for the mutant protein model. They are not strong supporting data.

They should have considered all these points before assuming the identified PLEC variant is associated with congenital insensitivity to pain. Although not perfect, the Standards and Guidelines for the Interpretation of Sequence Variants (Richards et al., 2015), that I suggest using, do not indicate pathogenicity. 

RESPONSE

Thank you for your comment. Congenital insensitivity to pain is an extremely rare condition. The estimated prevalence of the condition is one in a million (Kim et al, 2013; Schon et al, 2020; Cho et al, 2024). This condition is extremely rare and most of the previous reports were mainly based on single case series (Lischka et al., 2023). Our patients had congenital insensitivity to pain with dental anomalies, including tooth agenesis and root maldevelopment. This combination is unique and most likely overlooked in the previously reported patients.

As mentioned by the reviewer some algorithms predicted this variant to be benign (e.g. MutationTaster21, Uniprot, and ClinVar), however, it is important to note that these algorithms are largely based on heterozygosity–and heterozygosity may well show no phenotype (per other family members). We agree with your comment that the title was too definite. The title of the paper has been changed to “Homozygosity for a rare PLEC variant suggests a contributory role in congenital insensitivity to pain”.

The association of the PLEC variant and congenital insensitivity to pain is supported by

1) The rarity of the variant with its allele frequency is 0.000432 in the general population. This variant is not found in 1092 normal Thai controls (T-Rex). According to gnomAD, the homozygosity of this variant has never been reported.

2) The segregation of the variant with the phenotype (pedigree in Figure 1). The probability of complete segregation of variant and phenotype by chance is 1/2x8, or 0.00390625.

3) The amino acid residue Arg1347 is conserved across species. In addition the mutation is located in a domain not previously harboring any of the rare missense variants.

4) The variant is predicted to be deleterious (0.07), probably damaging (1.0), disease probabilty (0.832), and deleterious (-7.029) by SIFT (https://sift.bii.a-star.edu.sg/), PolyPhen-2 (http://genetics.bwh.harvard.edu/pph2/), SNPs&GO (http://snps.biofold.org/snps-and-go//snps-and-go.html), and PROVEAN (http://provean.jcvi.org/index.php), respectively. It is predicted as most likely to interfere with protein function (C65) and altered transmembrane protein (0.12) by Align-GVGD (http://agvgd.hci.utah.edu/) and MutPred (http://mutpred.mutdb.org/index.html), respectively.

5) No other rare variants are found in the known congenital insensitivity to pain genes, including ATL1, ATL3, GLA, KIF1A, NGF, DST, ELP1, PRDM12, RAB7A, RETREG1, ZFHX2, SPTLC2, TTR, WNK1, SCN11A, SPTLC1, MPV17, NAGLU, CLTCL1, FAAHP1, FLVCR1, SCN9A, and NTRK1.

6) The findings from whole mount in situ hybridization and immunohistochemistry support that the clinical features found in the patients, such as acro-osteolysis, corneal scars, dental anomalies, and underdeveloped maxilla and mandible are developmental defects caused by genetic variant in PLEC. In other words, these features are not exclusively neurogenic.

7) The association of the PLEC variant with congenital insensitivity to pain is strongly supported by the study of Plectin-deficient mice, which have been shown to have impaired pain sensitivity (Valencia et al., 2021).

Cho JH, Hwang S, Kwak YH, Yum MS, Seo GH, Koh JY, Ju YS, Yoon JH, Kang M, Do HS, Kim S, Kim GH, Bae H, Lee BH. Clinical and genetic characteristics of three patients with congenital insensitivity to pain with anhidrosis: Case reports and a review of the literature. Mol Genet Genomic Med. 2024 Apr;12(4):e2430. doi: 10.1002/mgg3.2430.

Haga N, Kubota M, Miwa Z. Epidemiology of hereditary sensory and autonomic neuropathy type IV and V in Japan. Am J Med Genet A. 2013 Apr;161A(4):871-4. doi: 10.1002/ajmg.a.35803.

Kim W, Guinot A, Marleix S, Chapuis M, Fraisse B, Violas P. Hereditary sensory and autonomic neuropathy type IV and orthopaedic complications. Orthop Traumatol Surg Res. 2013.

Valencia, R.G.; Mihailovska, E.; Winter, L.; Bauer, K.; Fischer, I.; Walko, G.; Jorgacevski, J.; Potokar, M.; Zorec, R.; Wiche, G. Plectin dysfunction in neurons leads to tau accumulation on microtubules affecting neuritogenesis, organelle trafficking, pain sensitivity and memory. Neuropathol Appl Neurobiol 2021, 47, 73-95, doi:10.1111/nan.12635.

  1. In section 3.2. Whole exome sequencing and bioinformatic analysis, the authors stated that the variant rs37225609 is predicted to be disease-causing (0.999999827907939), deleterious (0.07), and probably damaging (1) by Mutation Tasting, SIFT, and PolyPhen-2. They should add references for these prediction tools. I could not find Mutation Tasting. Did the authors use Mutation Taster instead, as also stated in line 134 of the ms? In this last case, https://www.genecascade.org/MutationTaster2021/#transcript the variant c.4039C>T; p.Arg1347Cys (rs372256096) is predicted as benign not as disease-causing.

RESPONSE

Thank you for the mistake. It should have been MutationTaster21. It has been corrected. References have been added to the manuscript as follows.

3.2. Whole exome sequencing and bioinformatic analysis

Whole exome and Sanger sequencing identified a homozygous PLEC variant (NM_000445.3: c.4039C>T; NP_000436.2: p.Arg1347Cys (rs372256096) in patients 1 and 2. The unaffected family members were heterozygous for the variant (Figure 1, 8A). This variant is predicted to be deleterious (0.07), probably damaging (1.0), disease probabilty (0.832), and deleterious (-7.029) by SIFT (https://sift.bii.a-star.edu.sg/), PolyPhen-2 (http://genetics.bwh.harvard.edu/pph2/), SNPs&GO (http://snps.biofold.org/snps-and-go//snps-and-go.html), and PROVEAN (http://provean.jcvi.org/index.php) , respectively. It is predicted as most likely to interfere with protein function (C65) and altered transmembrane protein (0.12) by Align-GVGD (http://agvgd.hci.utah.edu/) and MutPred (http://mutpred.mutdb.org/index.html), respectively. Additionally, the CADD (https://cadd.gs.washington.edu/) score of this variant is 24.8. The amino acid is conserved across species (Figure 8B). Bioinformation analysis revealed absence of the variants in the known congenital insensitivity to pain genes, including ATL1, ATL3, GLA, KIF1A, NGF, DST, ELP1, PRDM12, RAB7A, RETREG1, ZFHX2, SPTLC2, TTR, WNK1, SCN11A, SPTLC1, MPV17, NAGLU, CLTCL1, FAAHP1, FLVCR1, SCN9A, and NTRK1 [6, 7].

  1. The reported variant is not located in a particular isoform of the PLEC gene or in an exon where mutations are not found in other associated phenotypes.

RESPONSE

Thank you for your comment. Our study did not identify the isoform of PLECTIN. The c.4039C>T; p.Arg1347Cys variant in the PLEC gene is located in exon 31. The variants in this exon have not been reported to be associated with PLECTIN-associated syndromes. However, the variant we found clearly segregates with the phenotype in our study (pedigree in Figure 1), strongly supporting its causal role in the disease. In addition, our protein model analysis showed that the amino acid change resulting from the mutation has a significant impact on the protein structure. The p.Arg1347Cys variant is located in the SR9 region of the plakin domain (Figure 16A). The biological function of the spectrin repeats in the plakin domain is not fully understood but is thought to involve buffering of mechanical stress and/or mechanosensing. [Ortega et al., 2016]. Both of these functions involve unfolding and refolding in response to tension and require stability and dynamics to be finely tuned [Rief et al., 1999]. Although the replacement of Arg1347 with a cysteine as a result of the mutation would probably not be enough to abolish the structure in vitro, it leads to a loss of stabilizing contacts and hence may therefore affect the dynamics of SR9 under physiological mechanical stress. Moreover, the Arg137Cys mutation changes the stereochemistry of this surface region from charged to hydrophobic (see region circled in green in the figure below). Therefore, the PLEC variant may also perturb a subset of the many interactions that PLECTIN has with other proteins in the densely packed adhesion environments where it interconnects membrane-associated structures to the cytoskeleton. The impact on the SR9 (plakin region) is distinct from other missense variants in the rod domain, suggesting such variants would disrupt distinct functions of the protein.

Interestingly, AlphaFold strongly predicts homodimeric interactions for the SR7-SR9 fragment, involving charge-charge and hydrogen bond interactions for Arg1347. These interactions would be lost in the cysteine variant. In strong support, SR9 is positioned just before the rod domain, which is known to homodimerize [Ketema et al., 2015], and the asymmetric unit of the SR7-9 crystal structure (PDB id 5J1I) contains a dimeric arrangement as predicted by AlphaFold [Ortega et al., 2016].  

The figure of protein model and its legends are showed below.

Figure 15. Mutant protein model. A) Schematic drawing of human PLECTIN with domain boundary amino acids. SR: spectrin repeat; ABD: actin binding domain, composed of two calponin homology domains; PRD: plectin repeat domain. The location of the mutation Arg1347 in the ninth and last SR is indicated. The green square within the Plakin domain indicates the SH3 domain. B) Molecular model of the SR7-SR9 fragment, based on PDB 5jli. Arg1347 is shown as sphere model. C) The zoom figure shows the molecular environment of Arg1347. The side chain of the cysteine is shown with carbons in magenta and the sulfur in yellow. Side chains of other key residues are highlighted. D) AlphaFold predicts Sr7-9 homodimers. The two protein chains of the AlphaFold model are colored in gray and pale yellow. The model is superimposed on the crystallographic dimer of SR7-9 (PDB id 5j1i, cyan). Boxed region containing Arg1347 is shown enlarged in the close-up figure. E) Electrostatic surface of SR9, colored from blue (positively charged) over white (neutral) to red (negatively charged). Left and right views show the wild-type and mutant with position 1347 encircled (green). F) Predicted Aligned Error (PAE) matrix for SR7-9 homodimerization.

  1. The authors do not discuss enough the issue that this variant is associated with a clinical phenotype so different from the ones already described and do not give any possible explanation about it.

RESPONSE

Thank you for this comment. We have added the issue regarding the differences between the phenotypes found in our patients and those reported in the previously reported patients with PLEC variants. It is added in a new subheading as follows.

4.9 PLEC variant and its clinical phenotypes

     The clinical phenotypes found in our patients carrying a homozygous variant in the PLEC gene include congenital insensitivity to pain, corneal scarring, self-mutilation, auto-extraction of teeth, acro-osteolysis, and underdeveloped maxilla and mandible. The heterozygous variant c.4039C>T; p.Arg1347Cys in the PLEC gene is rare and its homozygous form has never been reported. As mentioned above, genetic variants in PLEC have been reported to be involved in plectin-associated diseases, including autosomal recessive generalized intermediate epidermolysis bullosa simplex 5D, autosomal dominant epidermolysis bullosa simplex 5A, Ogna type, autosomal recessive epidermolysis bullosa simplex 5B, with muscular dystrophy, autosomal recessive epidermolysis bullosa simplex with pyrolic atresia, and autosomal recessive limb-girdle muscular dystrophy-17 [9-11]. The location of the variant, epigenetic factors, modifying genes, and genetic background of the patients may account for the differences in phenotypes.

  1. It seems the authors considered only the autosomal recessive model. The authors stated that in order to test the hypothesis of the autosomal recessive model, 6,421 variants were extracted from the 177,299 variants that fitted the model and that the bioinformatic analysis revealed absence of the variants in the known congenital insensitivity to pain genes, including ATL1, ATL3, GLA, KIF1A, NGF, DST, ELP1, PRDM12, RAB7A, RETREG1, ZFHX2, SPTLC2, TTR, WNK1, SCN11A, SPTLC1, MPV17, NAGLU, CLTCL1, FAAHP1, FLVCR1, SCN9A, and NTRK1.Considering the pathogenic role of the variant found in PLEC is not adequately support, and the clinical phenotypes associate with mutations in PLEC are different from the one reported in the two Thai patients, the authors, in case they have not already done, should check again their exome findings. 

Are the known congenital insensitivity to pain genes adequately covered by probes in the SureSelect V6 UTR kit used?

Are there pathogenic/likely pathogenic variants in heterozygosity in these genes? Maybe the other allele is missing due to limitations of the technology, or maybe there is a different mechanism to be considered (i.e. see Complete paternal uniparental isodisomy for chromosome 1 revealed by mutation analyses of the TRKA (NTRK1) gene encoding a receptor tyrosine kinase for nerve growth factor in a patient with congenital insensitivity to pain with anhidrosis. Hum. Genet. 107: 205-209, 2000).

RESPONSE

Thank you for this comment and we appreciate the advice. The platform SureSelect V6 UTR kit which we used covered all known congenital insensitivity to pain genes. There were no rare variants in the known genes for congenital insensitivity to pain. We did not consider other mechanisms for the reasons described in RESPONSE # 2.

  1. Did the authors look for a de novo event or a dominant model? I understand the two sibs come from a consanguineous marriage, but the autosomal recessive model is not the only one being considered.

RESPONSE

Thank you for this comment. We looked at all possible modes of inheritance. There were no de novo variants that shared between both patients.

  1. They should have considered all these points before assuming the identified PLEC variant is associated with congenital insensitivity to pain. Although not perfect, the Standards and Guidelines for the Interpretation of Sequence Variants (Richards et al., 2015), that I suggest using, do not indicatepathogenicity. 

 RESPONSE

The response of this comment is in RESPONSE#2.

Minor comments

Figures 10, 12 and 16 should have a control to compare the findings.

RESPONSE

We added controls as suggested. Thank you.

In section 3.6.1. Plectin expression in the spinal dorsal horn, lines 343-345, “To investigate the potential involvement of PLECTIN in the congenital insensitivity to pain observed in our patients, an analysis of Plectin expression in the spinal dorsal horn of mouse embryos was conducted”. Please specify the day.

RESPONSE

We added Day E18.5. It is written as follows.

3.6.1. Plectin expression in the spinal dorsal horn

To investigate the potential involvement of PLECTIN in the congenital insensitivity to pain observed in our patients, an analysis of Plectin expression in the spinal dorsal horn of mouse embryos at E18.5 was conducted. Double staining with a neuronal cell marker, microtubule-associated protein 2 (MAP2), revealed that Plectin expression overlapped with MAP2 expression in the dorsal horn, indicating that Plectin is expressed in spinal dorsal horn neurons. (Figure 12A-H).

There are some inconsistencies for patient 2 in Fig. 6B and Fig. 8A, although the reported age is 19 for both figures, but the result is different, please adjust.

Figure 6. B) Patient 2. Corneal scar on the right eye. Left eye is unremarkable.

Figure 8. Patient 2 age 19 years. A) Corneal scars in both eyes

Furthermore, Fig. 6B can be moved to Fig. 8, so that there is one figure for each patient.

RESPONSE

Thank you for these comments. Patient 2 had corneal scar only on the right eye. We apologize for the mistake. We have corrected accordingly and combined the figures. Thank you for the advice.

Figure 7. Patient 2 age 19 years. A) Corneal scar in the right eye. Oral constriction as a result of scar contracture. B) Corneal scarring in the right eye. C) Loss of the tips of the index fingers. Thumbs appear unremarkable. D) Loss of the tips of all toes. Multiple scars are noted. E) Hand radiograph showing acro-osteolysis of the tips of second distal phalanges. F) Severe acro-osteolysis of the middle and distal phalanges of toes. G) Panoramic radiograph showing underdeveloped (narrow) maxilla and mandible. Absence of most permanent teeth. Mild taurodontism at the left and right maxillary second permanent molars (asterisks). Unseparated roots of the maxillary right permanent first molar (arrow). Alveolar socket as a result of a recent tooth extraction (star).

Figures of patient 1 at age 28 are also combined. Thank you for your suggestion.

Figure 5. Patient 1 at age 28 years. A, B) Severe corneal scarring in both eyes. The right eye is more severely affected than the left one. Multiple facial scars, bulbous nose, and oral contracture with scars are noted. C, D) Hands and feet. Loss of distal tips of fingers and toes. Thumbs have scars from previous traumas. No signs of loss of the tips. E, E) Radiographs of hands and feet showing acro-osteolysis of fingers and toes. Loss of the middle and distal phalanges. G) Panoramic radiograph at age 28 years showing underdeveloped maxilla and mandible. Absence of multiple permanent teeth is observed.

Reviewer 2 Report

Comments and Suggestions for Authors

The study presents intriguing findings but lacks some critical analyses.  Based on a preliminary investigation, the reported variant is probably benign. Moreover, it is located in a part of the protein where it doesn't have an impact. The fact that the variant is seen in three of the four individuals in the first generation of the family tree, where there doesn't seem to be any relatedness, lends further support to its benign character. At least 100 normal individuals should be evaluated as part of a population research to rule out the chance that the observed variant is common in the community under investigation. It is crucial to clarify its prevalence in the population notwithstanding the studies suggesting a possible influence in the animal model.

Author Response

Response to comments of Reviewer 2

The study presents intriguing findings but lacks some critical analyses. Based on a preliminary investigation, the reported variant is probably benign. Moreover, it is located in a part of the protein where it doesn't have an impact.

RESPONSE

Thank you for your comments. We have identified a homozygous rare variant in PLEC in two affected patients. This variant is rare with an allele frequency of 0.000432 according to gnomAD. This variant is not common in the Thai population as it is not found in 1092 healthy Thai controls. (T-REx database: https://trex.nbt.or.th; accessed on February 29, 2024). The unaffected family members (I-1, I-3, II-1, II-2, III-1, and III-3) are heterozygous for the variant. The probability that the unaffected members are heterozygous and the affected members are homozygous by chance is 1/2x8, or 0.00390625. The clear segregation of the variant and the phenotype in eight family members of the 3-generation pedigree strongly supports its causal role in the disease (Pedigree in Figure 1).

            Regarding the mutation predictions, this variant is predicted to be deleterious (0.07), probably damaging (1.0), disease probabilty (0.832), and deleterious (-7.029) by SIFT (https://sift.bii.a-star.edu.sg/), PolyPhen-2 (http://genetics.bwh.harvard.edu/pph2/), SNPs&GO (http://snps.biofold.org/snps-and-go//snps-and-go.html), and PROVEAN (http://provean.jcvi.org/index.php), respectively. It is predicted as most likely to disrupt with protein function (C65) and altered transmembrane protein (0.12) by Align-GVDB and MutaPred, respectively. The CADD (https://cadd.gs.washington.edu/) score is 24.8.

Regarding the location of the mutation, the mutated amino acid residue, Arg1347, is highly conserved across vertebrate species, suggesting its importance for PLEC structure and function. Arg1347 is located on the three-helical bundle structure of SR9, where it stabilizes the structure through polar contacts with nearby residues Y1343, E1293, and Q1296. The substitution with a cysteine is expected to disrupt these stabilizing bonds and decrease the stability of the SR9 unit. In addition, the Arg137Cys mutation changes the stereochemistry of this surface region from charged to hydrophobic (see region circled in green in the figure below). Therefore, the mutation may also disrupt a subset of the many interactions that plectin has with other proteins in the densely packed adhesion environments where it interconnects membrane-associated structures to the cytoskeleton.

Interestingly, AlphaFold strongly predicts homodimeric interactions for the SR7-SR9 fragment, which involve charge-charge and hydrogen bond interactions for Arg1347. These interactions would be lost in the cysteine variant. In strong support, SR9 is positioned just before the rod domain, which is known to homodimerize [Ketema et al.,  2015], and the asymmetric unit of the SR7-9 crystal structure (PDB id 5J1I) contains a dimeric arrangement as predicted by AlphaFold [Ortega et al., 2016].

The biological function of the spectrin repeats in the plakin domain is not fully understood, but is thought to involve buffering of mechanical stress and/or mechanosensing [Ortega et al.,  2016]. Both of these functions involve unfolding and refolding in response to tension and require stability and dynamics to be finely tuned [Rief et al., 1999]. Although the replacement of Arg1347 by a cysteine as a result of the mutation may not be sufficient to abolish the monomeric structure in vitro, it leads to a loss of stabilizing intra and intermolecular contacts which affect the interactions and dynamics of SR9 under physiological mechanical stress. Collectively, these lines of evidence strongly support that the mutation significantly impacts the protein function.

We have now updated the figure and its legends to clarify the important points as follows.

Figure 16. Mutant protein model. A. Schematic drawing of human PLECTIN with domain boundary amino acids. SR: spectrin repeat; ABD: actin binding domain, composed of two calponin homology domains; PRD: plectin repeat domain. The location of the mutation Arg1347 in the ninth and last SR is indicated. The green square within the Plakin domain indicates the SH3 domain. B.  Molecular model of the SR7-SR9 fragment, based on PDB 5jli. Arg1347 is shown as sphere model. C. The zoom figure shows the molecular environment of Arg1347. The side chain of the cysteine is shown with carbons in magenta and the sulfur in yellow. Side chains of other key residues are highlighted. D) AlphaFold predicts Sr7-9 homodimers. The two protein chains of the AlphaFold model are colored in gray and pale yellow. The model is superimposed on the crystallographic dimer of SR7-9 (PDB id 5j1i, cyan). Boxed region containing Arg1347 is shown enlarged in the close-up figure. E) Electrostatic surface of SR9, colored from blue (positively charged) over white (neutral) to red (negatively charged). Left and right views show the wild-type and mutant with position 1347 encircled (green). F) Predicted Aligned Error (PAE) matrix for SR7-9 homodimerization.

The location is also in contrast to other missense variants (in classic plectinopathies) which are mostly located in the C-terminal rod domain. There are increasing examples where different locations of variants in a gene/protein cause distinct phenotypes. So, we respectfully disagree that the variant is located in a region of the protein that has no impact. It is important to remember that missense variants clustered in the C-terminal rod domain may be a result of selection bias, since patients were selected based on consistency of phenotype.

The fact that the variant is seen in three of the four individuals in the first generation of the family tree, where there doesn't seem to be any relatedness, lends further support to its benign character. At least 100 normal individuals should be evaluated as part of a population research to rule out the chance that the observed variant is common in the community under investigation. It is crucial to clarify its prevalence in the population notwithstanding the studies suggesting a possible influence in the animal model.

RESPONSE

Thank you for this comment. The family we report is Muslim Thai. Individuals I-1, I-3, and I-4 carried a heterozygous variant c.4039C>T; p.Arg1347Cys. Individuals I-1 and I-3 are related as second cousins. Individual I-4 are not known to be related but his family has lived in the same village as individual I-3 for generations. A founder effect is possible.

It is crucial to clarify its prevalence in the population notwithstanding the studies suggesting a possible influence in the animal model.

RESPONSE

Thank you for your suggestion on doing the evaluation of this variant in at least 100 normal individuals. Our group is part of the nationwide group that did whole exome sequencing in 1092 normal Thais and this variant was NOT found in either heterozygous or homozygous forms (T-REx database: https://trex.nbt.or.th; accessed on February 29, 2024). Therefore, we can say that the congenital insensitivity to pain phenotype in two affected Thai siblings is extremely rare and caused by a novel homozygous variant in the PLEC gene. Our finding is supported by the study of Plectin-deficient mice, which have been shown to have impaired pain sensitivity (Valencia et al., 2021)

Valencia, R.G.; Mihailovska, E.; Winter, L.; Bauer, K.; Fischer, I.; Walko, G.; Jorgacevski, J.; Potokar, M.; Zorec, R.; Wiche, G. Plectin dysfunction in neurons leads to tau accumulation on microtubules affecting neuritogenesis, organelle trafficking, pain sensitivity and memory. Neuropathol Appl Neurobiol 2021, 47, 73-95, doi:10.1111/nan.12635.

Round 2

Reviewer 1 Report

Comments and Suggestions for Authors

The manuscript ijms-3013504, improved when the authors incorporated the suggestions.

However, there are still some arguments to be adjusted.

The authors state that the association between the PLEC variant and congenital insensitivity to pain is supported by some points. One is that according to GnomAD, the homozygosity of this variant has never been reported. In GnomAD v4.1.0, there are 2 homozygotes (1 East Asian). https://gnomad.broadinstitute.org/variant/8-143926870-G-A?dataset=gnomad_r4. How do they justify this finding?

The authors state in their response that “5) No other rare variants are found in the known congenital insensitivity to pain genes, including ATL1, ATL3, GLA, KIF1A, NGF, DST, ELP1, PRDM12, RAB7A, RETREG1, ZFHX2, SPTLC2, TTR, WNK1, SCN11A, SPTLC1, MPV17, NAGLU, CLTCL1, FAAHP1, FLVCR1, SCN9A, and NTRK1”. Still, it is unclear whether they checked for heterozygous rare pathogenic variants in such genes, considering the missing of the second allele. They should add in the ms that they checked this and looked at all possible modes of inheritance, considering the reported association between the PLEC variant and the congenital insensitivity to pain is only putative. This will strengthen the findings.

In the Patients’ description and in the Conclusion, the authors should clarify better whether the subjects in the first generation are related or come from the same village since they state that the identified variant is very rare. Is there a possible founder effect? Additional clarification will help support the variant's pathogenic role.

In Figure 11. Transmission electron micrography of the skin biopsy from patient 1 reveals fewer number of axons (arrows) than those of B) The normal skin (obtained from https://histolo- gyguide.com/EM-view/RD-060-peripheral-nerve/06-photo-1.html), please add A) and put before the control. Same in Figure 9, control at first.

In Figure 15. Mutant protein model., D) AlphaFold predicts Sr7-9 homodimers. The two protein chains of the AlphaFold model are colored in gray and pale yellow. The model is superimposed on the crystallographic dimer of SR7-9 (PDB id 5j1i, cyan). Boxed region containing Arg1347 is shown enlarged in the close-up figure. 4; Please indicate where the boxed region enlarged derives from the entire figure.

In the Conclusion;

- lines 655-657, the authors state: “Whole exome and Sanger sequencing identified a rare homozygous variant c.4039C>T; p.Arg1347Cys, located in the plakin domain of PLEC – a region not previously harboring missense variants in different plectinopathies [11]”. According to reference 11, the plakin domain includes missense variants in different plectinopathies. Please adjust.

- lines 662, 663 “This is the first report showing an association between a PLEC variant and congenital insensitivity to pain in humans”; again, this sentence is too strong, and the authors should instead use the term “putative association”.

Author Response

RESPONSE to comments of Reviewer 1

The manuscript ijms-3013504, improved when the authors incorporated the suggestions.

Thank you for your valuable comments and advice. You have made the paper better.

However, there are still some arguments to be adjusted.

The authors state that the association between the PLEC variant and congenital insensitivity to pain is supported by some points. One is that according to GnomAD, the homozygosity of this variant has never been reported. In GnomAD v4.1.0, there are 2 homozygotes (1 East Asian). https://gnomad.broadinstitute.org/variant/8-143926870-G-A?dataset=gnomad_r4. How do they justify this finding?

RESPONSE

Thank you for this comment. We did not mention this data, and we apologize for this. Yes, there are two homozygotes reported in gnomAD v4.1.0. Howeber, it is important to note that >60% of the new data included in the gnomAD v4.1.0 release when compared to gnomAD v2.1 derive from the UK Biobank. Specifically, 470,000 whole exomes were included from the UK Biobank, of which only ~25,000 are recognized as controls. Given most of the sequencing data for the UK Biobank were provided by companies conducting clinical sequencing, it is assumed that the remaining individuals have variable disease phenotypes. (https://gnomad.broadinstitute.org/news/2024-04-gnomad-v4-1/). We do not have
access to the UK Biobank phenotypic data to make further comment.     

            However, if there really are homozygotes without congenital insensitivity to pain phenotype, it will suggest that the presentation of congenital insensitivity to pain phenotype depends on other factors including epigenetic factors, environmental factors, the genetic background of the patients.

The manuscript was corrected as follows.

                        According to the gnomAD v2.1.1, the variant c.4039C>T; p.Arg1347Cys has a global allele             frequency of 0.0004320 in the general population with no reports of homo-zygotes (accessed from https://gnomad.broadinstitute.org on May 12, 2024). However, the gnomAD v4.1.0. reported its global allele frequency of 0.0001848 in the general population with report of two homozygotes (accessed from https://gnomad.broadinstitute.org on May 12, 2024). It is important to note that the patients included in gnomAD v4.1.0 were not all "normal" (https://gnomad.broadinstitute.org/stats). In fact, ~470,000 exomes from the UK Biobank, of which ~10% of samples are considered “controls”, were included (accessed from https://www.ukbiobank.ac.uk on May 14, 2024). Further investigation of the two individuals that are also homozygous for this PLEC variant is warranted to confirm the phenotypic association with pain insensitivity and dental anomalies. However, if there are truly normal homozygotes of this variant, it would suggest that the presentation of the congenital insensitivity to pain phenotype may depend on other factors, including epigenetic factors, environmental factors, and the genetic background of the patients.                                                                        .

The authors state in their response that “5) No other rare variants are found in the known congenital insensitivity to pain genes, including ATL1, ATL3, GLA, KIF1A, NGF, DST, ELP1, PRDM12, RAB7A, RETREG1, ZFHX2, SPTLC2, TTR, WNK1, SCN11A, SPTLC1, MPV17, NAGLU, CLTCL1, FAAHP1, FLVCR1, SCN9A, and NTRK1”. Still, it is unclear whether they checked for heterozygous rare pathogenic variants in such genes, considering the missing of the second allele. They should add in the ms that they checked this and looked at all possible modes of inheritance, considering the reported association between the PLEC variant and the congenital insensitivity to pain is only putative. This will strengthen the findings.

RESPONSE

Bioinformation analysis revealed absence of the variants in the known congenital insensitivity to pain genes, including ATL1, ATL3, GLA, KIF1A, NGF, DST, ELP1, PRDM12, RAB7A, RETREG1, ZFHX2, SPTLC2, TTR, WNK1, SCN11A, SPTLC1, MPV17, NAGLU, CLTCL1, FAAHP1, FLVCR1, SCN9A, and NTRK1 [6, 7]. We have searched for rare heterozygous and homozygous variants in these genes and considered all possible modes of inheritance, but have not found any other candidate variants for the congenital insensitivity to pain phenotype.

In the Patients’ description and in the Conclusion, the authors should clarify better whether the subjects in the first generation are related or come from the same village since they state that the identified variant is very rare. Is there a possible founder effect? Additional clarification will help support the variant's pathogenic role.

RESPONSE

Thank you for this comment. It is corrected as follows.

                        3.2. Whole exome sequencing and bioinformatic analysis

                                    Whole exome and Sanger sequencing identified a homozygous PLEC variant                               (NM_000445.3: c.4039C>T; NP_000436.2: p.Arg1347Cys (rs372256096) in patients 1 and 2.                             The unaffected family members were heterozygous for the variant (Figure 1, 8A). Of                         note, the patients’ family and the individual I-4 were unrelated but have lived in the                                same village for generations. Therefore, a founder effect is suspected.

In Figure 11. Transmission electron micrography of the skin biopsy from patient 1 reveals fewer number of axons (arrows) than those of B) The normal skin (obtained from https://histolo- gyguide.com/EM-view/RD-060-peripheral-nerve/06-photo-1.html), please add A) and put before the control. Same in Figure 9, control at first.

RESPONSE

Figure 9. A) Normal control. B) Patient 1. C) Patient 2. A) The normal skin control showing a normal thickness of the epidermis. Its dermal skin appendages and dermal nerve (arrows) are unremarkable. B, C) Patients’ epidermis showing normal thickness. Papillary and reticular dermis are unremarkable. Superficial vascular plexus is unremarkable. Nerve bundles are not seen. Sebaceous glands, and eccrine sweat glands are unremarkable. C) Mild superficial perivascular mononuclear cell infiltration.

Figure 11. Transmission electron micrography of A) Normal skin. B) The skin biopsy from patient 1 reveals fewer number of axons (arrows) than those of A) The normal skin (obtained from https://histologyguide.com/EM-view/RD-060-peripheral-nerve/06-photo-1.html).

In Figure 15. Mutant protein model., D) AlphaFold predicts Sr7-9 homodimers. The two protein chains of the AlphaFold model are colored in gray and pale yellow. The model is superimposed on the crystallographic dimer of SR7-9 (PDB id 5j1i, cyan). Boxed region containing Arg1347 is shown enlarged in the close-up figure. 4; Please indicate where the boxed region enlarged derives from the entire figure.

RESPONSE

Sorry it was our mistake in making figure. The box was lost and is now back in the figure.

In the Conclusion;

lines 655-657, the authors state: “Whole exome and Sanger sequencing identified a rare homozygous variant c.4039C>T; p.Arg1347Cys, located in the plakin domain of PLEC – a region not previously harboring missense variants in different plectinopathies [11]”. According to reference 11, the plakin domain includes missense variants in different plectinopathies. Please adjust.

RESPONSE

Thank you for this comment. Yes, to our knowledge there have been three other disease-associated missense variants reported in the plakin domain;. two at the N-terminal end of the domain (R323Q and R500G) and one towards the C-terminal end of the domain (R1029H) although still >300 amino acids from where the p.Arg1347Cys variant is located. Each of the three other missense variants in the plakin domain were found in patients that each had distinct phenotypes and, in one instance, a tissue-restricted clinical presentation (Castanon & Wiche, 2021). It is edited as follows.

                                    Whole exome and Sanger sequencing identified a rare homozygous variant                                              c.4039C>T; p.Arg1347Cys, which is located in the SR 9 region of the plakin                                      domain, a region not previously found to harbor pathogenic missense variants in                                     other plectinopathies. [11].

lines 662, 663 “This is the first report showing an association between a PLEC variant and congenital insensitivity to pain in humans”; again, this sentence is too strong, and the authors should instead use the term “putative association”.

RESPONSE

Thank you. We corrected it as follows.

                        This is the first report showing a putative association between a PLEC variant and                                    congenital insensitivity to pain in humans.

Round 3

Reviewer 1 Report

Comments and Suggestions for Authors

The ms has been improving. Just a few minor comments:

Please specify the human genome reference in line 302, chr8:145001038G>A; rs372256096.

In line 475, I'd suggest changing "normal homozygotes" to healthy homozygotes" for this variant.

Author Response

Responses to comments of Reviewer 1/3

The ms has been improving. Just a few minor comments:

Please specify the human genome reference in line 302, chr8:145001038G>A; rs372256096.

RESPONSE: Thank you so much for your kind comments and suggestions. They have improved the scientific quality of this paper. We genuinely thank you for this. We have corrected the manuscript as suggested.

3.2. Whole exome sequencing and bioinformatic analysis

 Whole exome and Sanger sequencing identified a homozygous PLEC  variant (NM_000445.3: c.4039C>T; NP_000436.2: p.Arg1347Cys;                                                       chr8:145001038G>A based on human reference sequence build GRCh37                                                 (hg19); rs372256096) in patients 1 and 2.

In line 475, I'd suggest changing "normal homozygotes" to healthy homozygotes" for this variant.

However, if there are truly healthy homozygotes of this variant, it would suggest that the presentation of the congenital insensitivity to pain phenotype may depend on other factors, including epigenetic factors, environmental factors, and the genetic background of the patients.
